# Equivariant Neural Simulators for Stochastic Spatiotemporal Dynamics

**Koen Minartz**[1]     **Yoeri Poels**[1,2]     **Simon Koop**[1]     **Vlado Menkovski**[1]

[1]Data and Artificial Intelligence Cluster, Eindhoven University of Technology
[2]Swiss Plasma Center, École Polytechnique Fédérale de Lausanne
{k.minartz, y.r.j.poels, s.m.koop, v.menkovski}@tue.nl

## Abstract

Neural networks are emerging as a tool for scalable data-driven simulation of high-dimensional dynamical systems, especially in settings where numerical methods are infeasible or computationally expensive. Notably, it has been shown that incorporating domain symmetries in deterministic neural simulators can substantially improve their accuracy, sample efficiency, and parameter efficiency. However, to incorporate symmetries in probabilistic neural simulators that can simulate stochastic phenomena, we need a model that produces *equivariant distributions over trajectories*, rather than equivariant function approximations. In this paper, we propose Equivariant Probabilistic Neural Simulation (EPNS), a framework for autoregressive probabilistic modeling of equivariant distributions over system evolutions. We use EPNS to design models for a stochastic n-body system and stochastic cellular dynamics. Our results show that EPNS considerably outperforms existing neural network-based methods for probabilistic simulation. More specifically, we demonstrate that incorporating equivariance in EPNS improves simulation quality, data efficiency, rollout stability, and uncertainty quantification. We conclude that EPNS is a promising method for efficient and effective data-driven probabilistic simulation in a diverse range of domains.

## 1   Introduction

The advent of fast and powerful computers has led to numerical simulations becoming a key tool in the natural sciences. For example, in computational biology, models based on probabilistic cellular automata are used to effectively reproduce stochastic cell migration dynamics [2, 13, 26, 51], and in physics, numerical methods for solving differential equations are used in diverse domains such as weather forecasting [4, 32], thermonuclear fusion [16, 55], and computational astrophysics [24, 52], amongst others. Recently, data-driven methods driven by neural networks have emerged as a complementary paradigm to simulation. These *neural simulators* are particularly useful when numerical methods are infeasible or impractical, for example due to computationally expensive simulation protocols or lack of an accurate mathematical model [9, 14, 30, 37, 41, 42].

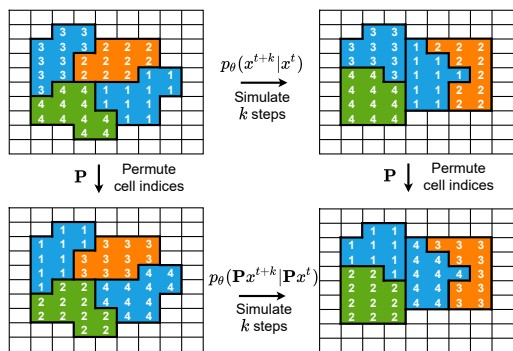

Figure 1: Schematic illustration of EPNS applied to stochastic cellular dynamics. EPNS models distributions over trajectories that are equivariant to permutations $\mathbf{P}$ of the cell indices, so that $p_\theta(\mathbf{P}x^{t+k}|\mathbf{P}x^t) = p_\theta(x^{t+k}|x^t)$.

37th Conference on Neural Information Processing Systems (NeurIPS 2023).

Notably, most neural simulators are deterministic: given some initial condition, they predict a single trajectory the system will follow. However, for many applications this formulation is insufficient. For example, random effects or unobserved exogenous variables can steer the system towards strongly diverging possible evolutions. In these scenarios, deterministic models are incentivized to predict an 'average case' trajectory, which may not be in the set of likely trajectories. Clearly, for these cases probabilistic neural simulators that can expressively model *distributions over trajectories* are required. Simultaneously, it has been shown that incorporating domain symmetries in model architectures can be highly beneficial for model accuracy, data efficiency, and parameter efficiency [10], all of which are imperative for a high-quality neural simulator. As such, both the abilities to perform probabilistic simulation and to incorporate symmetries are crucial to develop reliable data-driven simulators.

Accordingly, prior works have proposed methods for probabilistic simulation of dynamical systems, as well as for symmetry-aware simulation. For probabilistic spatiotemporal simulation, methods like neural stochastic differential equations (NSDEs) [20, 27], neural stochastic partial differential equations (NSPDEs) [40], Bayesian neural networks [34, 58], autoregressive probabilistic models [50], and Gaussian processes [59] have been successfully used. Still, these methods do not incorporate general domain symmetries into their model architectures, hindering their performance. In contrast, almost all works that focus on incorporating domain symmetries into neural simulation architectures are limited to the deterministic case [7, 18, 19, 31, 53, 54]. As such, these methods cannot simulate stochastic dynamics, which requires expressive modeling of distributions over system evolutions. A notable exception is [50], which proposes a method for probabilistic equivariant simulation, but only for two-dimensional data with rotation symmetry.

Although the above-mentioned works address the modeling of uncertainty or how to build equivariant temporal simulation models, they do not provide a general method for *equivariant stochastic simulation of dynamical systems*. In this work, we propose a framework for probabilistic neural simulation of spatiotemporal dynamics under equivariance constraints. Our main contributions are as follows:

- We propose Equivariant Probabilistic Neural Simulation (EPNS), an autoregressive probabilistic modeling framework for simulation of stochastic spatiotemporal dynamics. We mathematically and empirically demonstrate that EPNS produces single-step distributions that are equivariant to the relevant transformations, and that it employs these distributions to autoregressively construct equivariant distributions over the entire trajectory.

- We evaluate EPNS on two diverse problems: an n-body system with stochastic forcing, and stochastic cellular dynamics on a grid domain. For the first problem, we incorporate an existing equivariant architecture in EPNS. For the second, we propose a novel graph neural network that is equivariant to permutations of cell indices, as illustrated in Figure 1.

- We show that incorporating equivariance constraints in EPNS improves data efficiency, uncertainty quantification and rollout stability, and that EPNS outperforms existing methods for probabilistic simulation in the above-mentioned problem settings.

## 2 Background and related work

### 2.1 Equivariance

For a given group of transformations $G$, a function $f : X \to X$ is $G$-equivariant if

$$f(\rho(g)x) = \rho(g)f(x) \tag{1}$$

holds for all $g \in G$, where $\rho(g)$ denotes the group action of $g$ on $x \in X$ [10, 12]. In the context of probabilistic models, a conditional probability distribution $p(x|y, z)$ is defined to be $G$-equivariant with respect to $y$ if the following equality holds for all $g \in G$:

$$p(x|y, z) = p(\rho(g)x|\rho(g)y, z). \tag{2}$$

The learning of such equivariant probability distributions has been employed for modeling symmetrical density functions, for example for generating molecules in 3D (Euclidean symmetry) [25, 43, 57] and generating sets (permutation symmetry) [6, 28]. However, different from our work, these works typically consider time as an internal model variable, and use the equivariant probability distribution as a building block for learning invariant probability density functions over static data points by starting from a base distribution that is invariant to the relevant transformations. For example, each

3D orientation of a molecule is equally probable. Another interesting work is [17], which considers equivariant learning of stochastic fields. Given a set of observed field values, their methods produce equivariant distributions over interpolations between those values using either equivariant Gaussian Processes or Steerable Conditional Neural Processes. In contrast, our goal is to produce equivariant distributions over temporal evolutions, given only the initial state of the system.

## 2.2 Simulation with neural networks

Two commonly used backbone architectures for neural simulation are graph neural networks for operating on arbitrary geometries [1, 3, 9, 11, 18, 29, 36, 42, 56] and convolution-based models for regular grids [14, 30, 48]. Recently, the embedding of symmetries beyond permutation and translation equivariance has become a topic of active research, for example in the domains of fluid dynamics and turbulence modeling [31, 53, 54], climate science [7], and various other systems with Euclidean symmetries [18, 19, 44]. Although these works stress the relevance of equivariant simulation, they do not apply to the stochastic setting due to their deterministic nature.

Simultaneously, a variety of methods have been proposed for probabilistic simulation of dynamical systems. Latent neural SDEs [27] and ODE$^2$VAE [58] simulate dynamics by probabilistically evolving a (stochastic) differential equation in latent space. Alternatively, methods like Bayesian neural networks (BNNs) and Gaussian processes (GPs) have also been applied to probabilistic spatiotemporal simulation [34, 59]. Further, NSPDEs are able to effectively approximate solutions of stochastic partial differential equations [40]. Due to their probabilistic nature, all of these methods are applicable to stochastic simulation of dynamical systems, but they do not incorporate general domain symmetries into their models. Finally, closely related to our work, [50] addresses the problem of learning equivariant distributions over trajectories in the context of multi-agent dynamics. However, their method is limited to two spatial dimensions, assumes a Gaussian one-step distribution, and only takes the symmetry group of planar rotations into account. In contrast, we show how general equivariance constraints can be embedded in EPNS, and that this leads to enhanced performance in two distinct domains.

## 3 Method

### 3.1 Modeling framework

**Problem formulation.** At each time $t$, the system's state is denoted as $x^t$, and a sequence of states is denoted as $x^{0:T}$. We presuppose that we have samples from a ground-truth distribution $p^*(x^{0:T})$ for training, and for inference we assume that $x^0$ is given as a starting point for the simulation. In this work, we assume Markovian dynamics, although our approach can be extended to non-Markovian dynamics as well. Consequently, the modeling task boils down to maximizing Equation 3:

$$\mathbb{E}_{x^{0:T} \sim p^*} \mathbb{E}_{t \sim U\{0,...,T-1\}} \log p_\theta(x^{t+1}|x^t). \tag{3}$$

As such, we can approximate $p^*(x^{1:T}|x^0)$ by learning $p_\theta(x^{t+1}|x^t)$ and applying it autoregressively. Furthermore, when we know that the distribution $p^*(x^{1:T}|x^0)$ is equivariant to $\rho(g)$, we want to guarantee that $p_\theta(x^{1:T}|x^0)$ is equivariant to these transformations as well. This means that

$$p_\theta(x^{1:T}|x^0) = p_\theta(\rho(g)x^{1:T}|\rho(g)x^0) \tag{4}$$

should hold, where $\rho(g)x^{1:T}$ simply means applying $\rho(g)$ to all states in the sequence $x^{1:T}$.

**Generative model.** The model $p_\theta(x^{t+1}|x^t)$ should adhere to three requirements. First, since our goal is to model equivariant distributions over trajectories in diverse domains, the type of model used for $p_\theta$ should be amenable to incorporating general equivariance constraints. Second, we argue that generative models with iterative sampling procedures, although capable of producing high-quality samples [15, 46], are impractical in this setting as we need to repeat the sampling procedure over potentially long trajectories. Third, the model should be able to simulate varying trajectories for the same initial state, especially in the case of sensitive chaotic systems, so good mode coverage is vital.

As such, we model $p_\theta(x^{t+1}|x^t)$ as $\int_z p_\theta(x^{t+1}|z, x^t)p_\theta(z|x^t)dz$ by using a latent variable $z$, following the conditional Variational Autoencoder (CVAE) framework [47]. Specifically, to generate $x^{t+1}$, $x^t$ is first processed by a *forward model* $f_\theta$, producing an embedding $h^t = f_\theta(x^t)$. If there is little

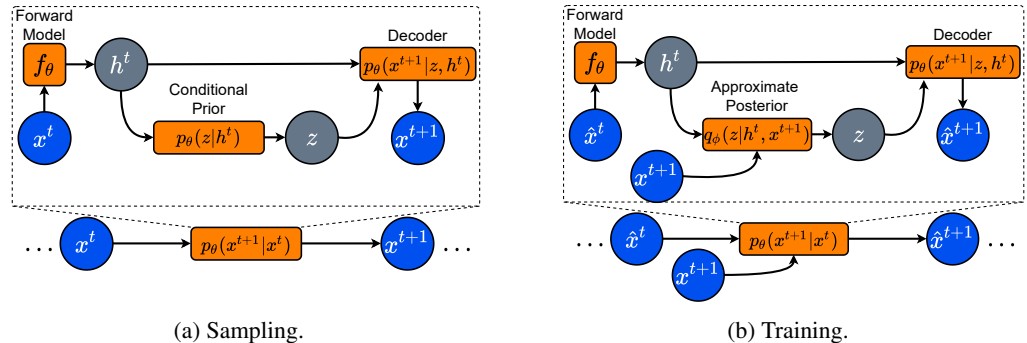

(a) Sampling.  (b) Training.

Figure 2: EPNS model overview.

variance in $p^*(x^{t+1}|x^t)$, $h^t$ contains almost all information for generating $x^{t+1}$. On the other hand, if the system is at a point where random effects can result in strongly bifurcating trajectories, $h^t$ only contains implicit information about possible continuations of the trajectories. To enable the model to simulate the latter kind of behavior, the embedding $h^t$ is subsequently processed by the conditional prior distribution. Then, $z \sim p_\theta(z|h^t)$ is processed in tandem with $h^t$ by the decoder, producing a distribution $p_\theta(x^{t+1}|z, h^t)$ over the next state. The sampling process is summarized in Figure 2a.

**Incorporating symmetries.**  Using the approach described above, we can design models that can simulate trajectories by iteratively sampling from $p_\theta(x^{t+1}|x^t)$. Now, the question is how we can ensure that $p_\theta(x^{1:T}|x^0)$ respects the relevant symmetries. In Appendix A, we prove Lemma 1:

**Lemma 1.** *Assume we model $p_\theta(x^{1:T}|x^0)$ as described in Section 3.1. Then, $p_\theta(x^{1:T}|x^0)$ is equivariant to linear transformations $\rho(g)$ of a symmetry group $G$ in the sense of Definition 2 if:*

(a) *The forward model is $G$-equivariant: $f_\theta(\rho(g)x) = \rho(g)f_\theta(x)$;*

(b) *The conditional prior is $G$-invariant or $G$-equivariant: $p_\theta(z|\rho(g)h^t) = p_\theta(z|h^t)$, or $p_\theta(\rho(g)z|\rho(g)h^t) = p_\theta(z|h^t)$;*

(c) *The decoder is $G$-equivariant with respect to $h^t$ (for an invariant conditional prior), or $G$-equivariant with respect to both $h^t$ and $z$ (for an equivariant conditional prior): $p_\theta(x^{t+1}|z, h^t) = p_\theta(\rho(g)x^{t+1}|z, \rho(g)h^t)$, or $p_\theta(x^{t+1}|z, h^t) = p_\theta(\rho(g)x^{t+1}|\rho(g)z, \rho(g)h^t)$.*

The consequence of Lemma 1 is that, given an equivariant neural network layer, we can stack these layers to carefully parameterize the relevant model distributions, guaranteeing equivariant distributions *over the entire trajectory*. To understand this, let us consider the case where the conditional prior is $G$-invariant, assuming two inputs $x^t$ and $\rho(g)x^t$. First, applying $f_\theta$ to $x^t$ yields $h^t$, and since $f_\theta$ is $G$−equivariant, applying it to $\rho(g)x^t$ yields $\rho(g)h^t$. Second, due to the invariance of $p_\theta(z|h^t)$, both $h^t$ and $\rho(g)h^t$ lead to exactly the same distribution over $z$. Third, since $z \sim p_\theta(z|h^t)$ is used to parameterize the decoder $p_\theta(x^{t+1}|z, h^t)$, which is equivariant with respect to $h^t$, the model's one-step distribution $p_\theta(x^{t+1}|x^t)$ is equivariant as well. Finally, autoregressively sampling from the model inductively leads to an equivariant distribution over the entire trajectory.

**Invariant versus equivariant latents.**  When $p_\theta(z|h^t)$ is equivariant, the latent variables are typically more local in nature, whereas they are generally more global in the case of an invariant conditional prior. For example, for a graph domain, we may design $p_\theta(z|h^t)$ to first perform a permutation-invariant node aggregation to obtain a global latent variable, or alternatively to equivariantly map each node embedding to a node-wise distribution over $z$ to obtain local latent variables. Still, both design choices result in an equivariant model distribution $p_\theta(x^{1:T}|x^0)$.

## 3.2  Training procedure

Following the VAE framework [23, 39], an approximate posterior distribution $q_\phi(z|x^t, x^{t+1})$ is learned to approximate the true posterior $p_\theta(z|x^t, x^{t+1})$. During training, $q_\phi$ is used to optimize the

Evidence Lower Bound (ELBO) on the one-step log-likelihood:

$$\log p_\theta(x^{t+1}|x^t) \geq \mathbb{E}_{q_\phi(z|x^t, x^{t+1})}\left[p_\theta(x^{t+1}|z, x^t)\right] - KL\left[q_\phi(z|x^t, x^{t+1})||p_\theta(z|x^t)\right]. \quad (5)$$

In our case, $q_\phi$ takes both the current and next timestep as input to infer a posterior distribution over the latent variable $z$. Naturally, we parameterize $q_\phi$ to be invariant or equivariant with respect to both $x^t$ and $x^{t+1}$ for an invariant or equivariant conditional prior $p_\theta(z|x^t)$ respectively.

A well-known challenge for autoregressive neural simulators concerns the accumulation of error over long rollouts [9, 48]: imperfections in the model lead to its output deviating further from the ground-truth for increasing rollout length $k$, producing a feedback loop that results in exponentially increasing errors. In the stochastic setting, we observe a similar phenomenon: $p_\theta(x^{t+1}|x^t)$ does not fully match $p^*(x^{t+1}|x^t)$, resulting in a slightly biased one-step distribution. Empirically, we see that these biases accumulate, leading to poor coverage of $p^*(x^{t+k}|x^t)$ for $k \gg 1$, and miscalibrated uncertainty estimates. Fortunately, we found that a heuristic method can help to mitigate this issue. Similar to pushforward training [9], in *multi-step* training we unroll the model for more than one step during training. However, iteratively sampling from $p_\theta(x^{t+1}|x^t)$ can lead to the simulated trajectory diverging from the ground-truth sample. Although this behavior is desired for a stochastic simulation model, it ruins the training signal. Instead, we iteratively sample a reconstruction $\hat{x}^{t+1}$ from the model, where $\hat{x}^{t+1} \sim p_\theta(x^{t+1}|\hat{x}^t, z)$ and $z \sim q_\phi(z|x^{t+1}, \hat{x}^t)$ as illustrated in Figure 2b. Using the posterior distribution over the latent space steers the model towards roughly following the ground truth. Additionally, instead of only calculating the loss at the last step of the rollout as done in [9], we calculate and backpropagate the loss at every step.[1]

## 4 Applications and model designs

### 4.1 Celestial dynamics with stochastic forcing

**Problem setting.** For the first problem, we consider three-dimensional celestial mechanics, which are traditionally modeled as a deterministic n-body system. However, in some cases the effects of dust distributed in space need to be taken into account, which are typically modeled as additive Gaussian noise on the forcing terms [5, 33, 45]. In this setting, the dynamics of the system are governed by the following SDE:

$$dx_i = v_i dt \qquad\qquad \forall i \in [n] \quad (6)$$

$$dv_i = \sum_{j \neq i} \frac{Gm_j(x_j - x_i)}{||x_j - x_i||^3}dt + \sigma(x)dW_t \qquad\qquad \forall i \in [n] \quad (7)$$

where $\sigma(x) \in \mathbb{R}$ is 1% of the magnitude of the deterministic part of the acceleration of the $i$'th body. We approximate Equation 6 with the Euler-Maruyama method to generate the data. Because of the chaotic nature of n-body systems, the accumulation of the noising term $\sigma(x)dW_t$ leads to strongly diverging trajectories, posing a challenging modeling problem.

**Symmetries and backbone architecture.** We model the problem as a fully connected geometric graph in which the nodes' coordinates evolve over time. As such, the relevant symmetries are (1) permutation equivariance of the nodes, and (2) E(n)-equivariance. We use a frame averaging GNN model as detailed in Section 3.3 of [38] as backbone architecture, denoted as `FA-GNN`, since it has demonstrated state-of-the-art performance in n-body dynamics prediction [38]. Using `FA-GNN`, we can parameterize functions that are invariant or equivariant to Euclidean symmetries. We also tried the EGNN [44], but during preliminary experimentation we found that training was relatively unstable in our setting, especially combined with multi-step training. The `FA-GNN` suffered substantially less from these instabilities.

**Model design.** Before explaining the individual model components, we first present the overall model structure, also depicted in Figure 3a:

- Forward model $f_\theta$: $\texttt{FA-GNN}_{\text{equivariant}}(x^t)$
- Conditional prior $p_\theta(z|h^t)$: $\mathcal{N}\left(\mu^{\text{prior}}(h^t), \sigma^{\text{prior}}(h^t)\right)$
- Decoder $p_\theta(x^{t+1}|z, h^t)$: $\mathcal{N}\left(\mu^{\text{decoder}}(z, h^t), \sigma^{\text{decoder}}(z, h^t)\right)$

---

[1]However, just as in [9], the gradients are only backpropagated through a single autoregressive step.

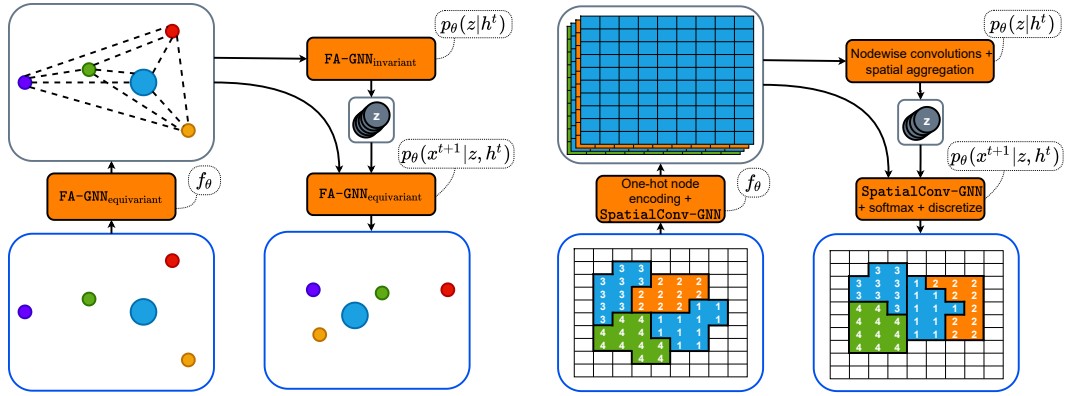

(a) Model structure for celestial dynamics.  (b) Model structure for cellular dynamics.

Figure 3: Schematic overviews of model designs.

As $f_\theta$ needs to be equivariant, we parameterize it with the equivariant version of the `FA-GNN`, denoted as `FA-GNN`$_{\text{equivariant}}$. Recall that $p_\theta(z|h^t)$ can be either invariant or equivariant. We choose for node-wise latent variables that are invariant to Euclidean transformations. As such, we parameterize $\mu^{\text{prior}}$ and $\sigma^{\text{prior}}$ with `FA-GNN`$_{\text{invariant}}$. Finally, $p_\theta(x^{t+1}|z, h^t)$ needs to be equivariant, which we achieve with a normal distribution where the parameters $\mu^{\text{decoder}}$ and $\sigma^{\text{decoder}}$ are modeled with `FA-GNN`$_{\text{equivariant}}$ and `FA-GNN`$_{\text{invariant}}$ respectively. More details can be found in Appendix C.1.

### 4.2 Stochastic cellular dynamics

**Problem setting.** The second problem considers spatiotemporal migration of biological cells. The system is modeled by the Cellular Potts model (CPM), consisting of a regular lattice $L$, a set of cells $C$, and a time-evolving function $x : L \to C$ [13]. Furthermore, each cell has an associated type $\tau(c)$. See also Figures 1 and 3b for illustrations. Specifically, we consider an extension of the cell sorting simulation, proposed in [13], which is a prototypical application of the CPM. In this setting, the system is initialized as a randomly configured culture of adjacent cells. The dynamics of the system are governed by the Hamiltonian, which in the cell sorting case is defined as follows:

$$H = \underbrace{\sum_{l_i, l_j \in \mathcal{N}(L)} J\left(x(l_i), x(l_j)\right)\left(1 - \delta_{x(l_i), x(l_j)}\right)}_{\text{contact energy}} + \underbrace{\sum_{c \in C} \lambda_V\left(V(c) - V^*(c)\right)^2}_{\text{volume constraint}}, \qquad (8)$$

where $\mathcal{N}(L)$ is the set of all adjacent lattice sites in $L$, $J\left(x(l_i), x(l_j)\right)$ is the contact energy between cells $x(l_i)$ and $x(l_j)$, and $\delta_{x,y}$ is the Kronecker delta. Furthermore, $C$ is the set of all cells in the system, $V(c)$ is the volume of cell $c$, $V^*(c)$ is the target volume of cell $c$, and $\lambda_V$ is a Lagrange multiplier. To evolve the system, an MCMC algorithm is used, which randomly selects an $l_i \in L$, and proposes to change $x(l_i)$ to $x(l_j)$, where $(l_i, l_j) \in \mathcal{N}(L)$. The change is accepted with probability $\min(1, e^{-\Delta H/T})$, where $T$ is the *temperature* parameter of the model. The values of the model parameters can be found in Appendix B.

**Symmetries and backbone architecture.** As the domain $L$ is a regular lattice, one relevant symmetry is shift equivariance. Further, since $x$ maps to a *set* of cells $C$, the second symmetry we consider is equivariance to permutations of $C$; see Figure 1 for an intuitive schematic. To respect both symmetries, we propose a novel message passing graph neural network architecture. We first transform the input $x$ to a one-hot encoded version, such that $x$ maps all lattice sites to a one-hot vector of cell indices along the channel axis. We then consider each channel as a separate node $h_i$. Subsequently, message passing layers operate according to the following update equation:

$$h_i^{l+1} = \psi^l\left(h_i^l, \bigoplus_{j \in \mathcal{N}(i)} \phi^l(h_j^l)\right), \qquad (9)$$

where $\bigoplus$ is a permutation-invariant aggregation function, performed over neighboring nodes. Usually, $\psi^l$ and $\phi^l$ would be parameterized with MLPs, but since $h_i^l$ lives on a grid domain, we choose both functions to be convolutional neural nets. We refer to this architecture as `SpatialConv-GNN`.

**Model design.**   We again start by laying out the overall model structure, also shown in Figure 3b:

- Forward model $f_\theta$: `SpatialConv-GNN`$(x^t)$
- Conditional prior: $p_\theta(z|h^t)$: $\mathcal{N}\left(\mu^{\text{prior}}(h^t), \sigma^{\text{prior}}(h^t)\right)$
- Decoder: $p_\theta(x^{t+1}|z, h^t)$: $\mathcal{C}\left(\pi^{\text{decoder}}(z, h^t)\right)$

Since $f_\theta$ is parameterized by a `SpatialConv-GNN`, it is equivariant to translations and permutations. At any point in time, each cell is only affected by itself and the directly adjacent cells, so we design $p_\theta(z|h^t)$ to be equivariant with respect to permutations, but invariant with respect to translations; concretely, we perform alternating convolutions and spatial pooling operations for each node $h_i$ to infer distributions over a latent variable $z$ for each cell. Finally, the decoder is parameterized by $\pi^{\text{decoder}}(z, h^t)$, a `SpatialConv-GNN` which equivariantly maps to the parameters of a pixel-wise categorical distribution over the set of cells $C$. More details can be found in Appendix C.2.

# 5   Experiments

## 5.1   Experiment setup

**Objectives.**   Our experiments aim to **a)** assess the effectiveness of EPNS in terms of simulation faithfulness, calibration, and stability; **b)** empirically verify the equivariance property of EPNS, and investigate the benefits of equivariance in the context of probabilistic simulation; and **c)** compare EPNS to existing methods for probabilistic simulation of dynamical systems. We report the results conveying the main insights in this section; further results and ablations are reported in Appendix E.

**Data.**   For both problem settings, we generate training sets with 800 trajectories and validation and test sets each consisting of 100 trajectories. The length of each trajectory is 1000 for the celestial dynamics case, saved at a temporal resolution of $dt = 0.01$ units per step. We train the models to predict a time step of $\Delta t = 0.1$ ahead, resulting in an effective rollout length of 100 steps for a full trajectory. Similar to earlier works [38, 44, 59], we choose $n = 5$ bodies for our experiments. For the cellular dynamics case, we use trajectories with $|C| = 64$ cells. The length of each trajectory is 59 steps, where we train the models to predict a single step ahead. Additionally, we generate test sets of 100 trajectories starting from the same initial condition, which are used to assess performance regarding uncertainty quantification.

**Implementation and baselines.**   The EPNS implementations are based on the designs explained in Section 4. The maximum rollout lengths used for multi-step training are 16 steps (celestial dynamics) and 14 steps (cellular dynamics). We use a linear KL-annealing schedule [8], as well as the 'free bits' modification of the ELBO as proposed in [22] for the cellular dynamics data. For celestial dynamics, we compare to neural SDEs (NSDE) [27] and interacting Gaussian Process ODEs (iGPODE) [59], as these methods can probabilistically model dynamics of (interacting) particles. For cellular dynamics, we compare to ODE$^2$VAE [58], as it probabilistically models dynamics on a grid domain. For each baseline, we search over the most relevant hyperparameters, considering at least seven configurations per baseline. We also compare to a non-equivariant counterpart of EPNS, which we will refer to as PNS. All models are implemented in PyTorch [35] and trained on a single NVIDIA A100 GPU. Further details on the baseline models can be found in Appendix D. Our code is available at `https://github.com/kminartz/EPNS`.

## 5.2   Results

**Qualitative results.**   Figure 4 shows a sample from the test set, as well as simulations produced by neural simulators starting from the same initial condition $x^0$ for both datasets. More samples are provided in Appendix E. For celestial dynamics, both EPNS and iGPODE produce a simulation that looks plausible, as all objects follow orbiting trajectories with reasonable velocities. As expected, the simulations start to deviate further from the ground truth as time proceeds, due to the accumulation

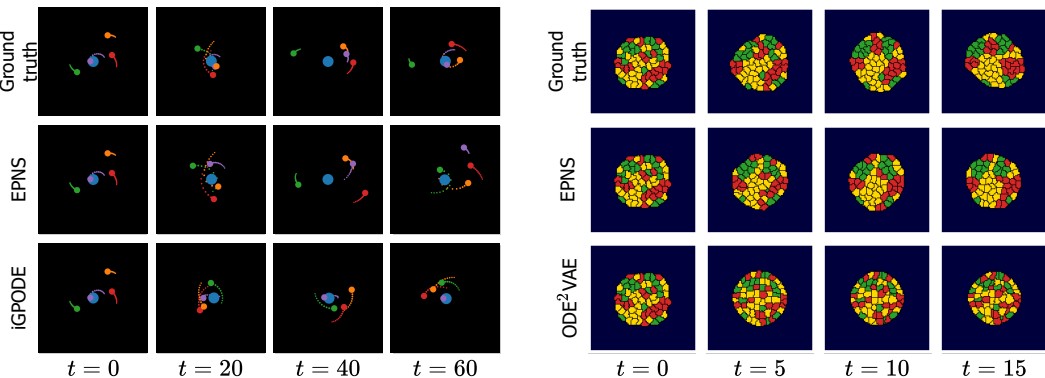

Figure 4: Qualitative results for celestial dynamics (left) and cellular dynamics (right).

Table 1: Comparison of EPNS to related work. For celestial dynamics, the kinetic energy of the system is used to calculate $D_{KS}$, while for celullar dynamics, we use the number of cell clusters of the same type. We report the mean±standard error over three independent training runs.

| | **Celestial dynamics** | | | | **Cellular dynamics** | | |
| Model | LL ↑ $\cdot 10^3$ | $D_{KS}(KE)$ ↓ t=50 | t=100 | Model | LL ↑ $\cdot 10^4$ | $D_{KS}$(#clusters) ↓ t=30 | t=45 |
|---|---|---|---|---|---|---|---|
| NSDE [27] | -5.0±0.0 | 0.80±0.01 | 0.65±0.03 | ODE$^2$VAE [58] | -30.1±0.0 | 0.98±0.01 | 0.96±0.04 |
| iGPODE [59] | -8.5±0.1 | 0.42±0.07 | 0.57±0.02 | | | | |
| PNS (ours) | **10.9**±0.4 | 0.61±0.18 | 0.20±0.06 | PNS (ours) | -16.4±0.3 | 0.70±0.10 | 0.77±0.05 |
| EPNS (ours) | **10.8**±0.1 | **0.14**±0.03 | **0.14**±0.04 | EPNS (ours) | **-5.9**±0.1 | **0.58**±0.09 | **0.58**±0.05 |

of randomness. In the cellular dynamics case, the dynamics produced by EPNS look similar to the ground truth, as cells of the same type tend to cluster together. In contrast, ODE$^2$VAE fails to capture this behavior. Although cells are located at roughly the right location for the initial part of the trajectory, the clustering and highly dynamic nature of the cells' movement and shape is absent.

**Comparison to related work.** We consider two metrics to compare to baselines: the first is the ELBO on the log-likelihood on the test set, denoted as LL. LL is a widely used metric in the generative modeling community for evaluating probabilistic models, expressing goodness-of-fit on unseen data, see for example [15, 22, 23, 59]. Still, the calculation of this bound differs per model type, and it does not measure performance in terms of new sample quality. This is why the second metric, the Kolmogorov-Smirnov (KS) test statistic $D_{KS}$, compares empirical distributions of newly sampled simulations to ground truth samples. Specifically, $D_{KS}$ is defined as $D_{KS}(y) = \max_y |F^*(y) - F_\theta(y)|$, the largest absolute difference between the ground truth and model empirical cumulative distribution functions $F^*$ and $F_\theta$. $D_{KS} = 0$ corresponds to identical samples, whereas $D_{KS} = 1$ corresponds to no distribution overlap at all. We opt for this metric since we have explicit access to the ground-truth simulator, and as such we can calculate $D_{KS}$ over 100 simulations starting from a fixed $x^0$ in order to assess how well EPNS matches the ground-truth distribution over possible trajectories. The results are shown in Table 1. EPNS performs comparative to or better than the other methods in terms of LL, indicating a distribution that has high peaks around the ground truth samples. In terms of $D_{KS}$, EPNS performs best, meaning that simulations sampled from EPNS also match better to the ground truth in terms of the examined properties of Table 1.

**Uncertainty quantification.** In addition to Tables 1 and 3, we further investigate how distributions produced by EPNS, PNS, and EPNS trained with one step training (EPNS-one step) align with the ground truth. Figure 5 shows how properties of 100 simulations, all starting from the same $x^0$, evolve over time. In the case of celestial dynamics, the distribution over potential energy produced by EPNS closely matches the ground truth, as shown by the overlapping shaded areas. In contrast, EPNS-one step effectively collapses to a deterministic model. Although PNS produces a distribution that overlaps with the ground truth, it doesn't match as closely as EPNS. For the cellular dynamics

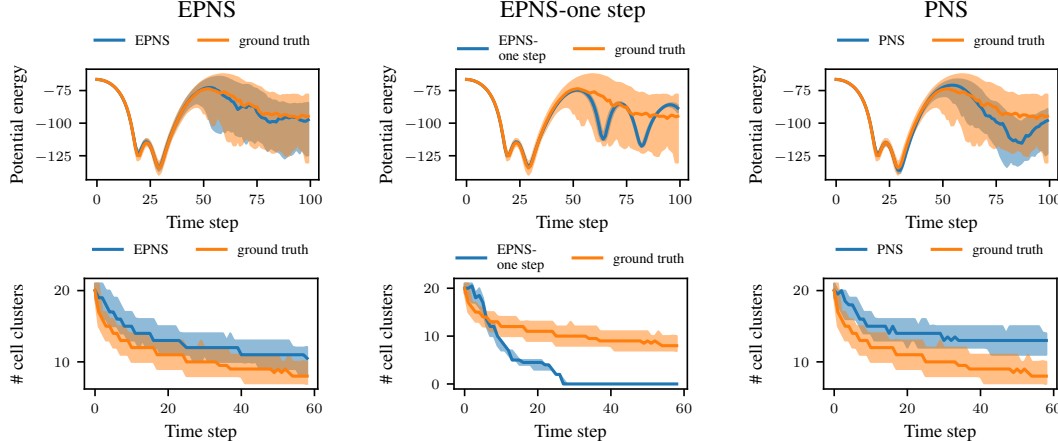

Figure 5: Distributions of the potential energy in a system (top row, celestial dynamics) and the number of cell clusters of the same type (bottom row, cellular dynamics) over time. The shaded area indicates the $(0.1, 0.9)$-quantile interval of the observations; the solid line indicates the median value.

case, the distribution produced by EPNS is slightly off compared to the ground truth, but still matches considerably better than EPNS-one step and PNS. These results indicate that both equivariance and multi-step training improve uncertainty quantification for these applications.

**Verification of equivariance.** To empirically verify the equivariance property of EPNS, we sample ten simulations for each $x^0$ in the test set, leading to 1000 simulations in total, and apply a transformation $\rho(g)$ to the final outputs $x^T$. $\rho(g)$ is a random rotation and random permutation for celestial dynamics and cellular dynamics respectively. Let us refer to the transformed outcomes as distribution 1. We then repeat the procedure, but now transform the *inputs* $x^0$ by the same transformations before applying EPNS, yielding outcome distribution 2. For an equivariant model, distributions 1 and 2 should be statistically indistinguishable. We test this by comparing attributes of a single randomly chosen body or cell using a two-sample KS-test. We also show results of PNS to investigate if equivariance is learned by a non-equivariant model. The results are shown in Table 2. In all cases, distributions 1 and 2 are statistically indistinguishable for EPNS ($p \gg 0.05$), but not for PNS ($p \ll 0.05$). As such, EPNS indeed models equivariant distributions over trajectories, while PNS does not learn to be equivariant.

Table 2: KS-test results for equivariance verification.

| | Celestial Dynamics | | | | | | Cellular Dynamics | |
|---|---|---|---|---|---|---|---|---|
| | x-coordinate | | y-coordinate | | z-coordinate | | distance traveled | |
| Model | $D_{KS}$ | p-value | $D_{KS}$ | p-value | $D_{KS}$ | p-value | $D_{KS}$ | p-value |
| PNS | 0.09 | 0.00 | 0.22 | 0.00 | 0.09 | 0.00 | 0.503 | 0.00 |
| EPNS | 0.04 | 0.37 | 0.02 | 0.99 | 0.02 | 0.98 | 0.036 | 0.536 |

**Data efficiency.** We now examine the effect of equivariance on data efficiency. Table 3 shows the ELBO of both EPNS and PNS when trained on varying training set sizes. In the celestial dynamics case, the ELBO of EPNS degrades more gracefully when shrinking the training set compared to PNS. For cellular dynamics, the ELBO of EPNS does not worsen much at all when reducing the dataset to only 80 trajectories, while the ELBO of PNS deteriorates rapidly. These results demonstrate that EPNS is more data efficient due to its richer inductive biases.

Table 3: Test set ELBO values ($\cdot 10^3$) for varying amounts of training samples.

| # Training samples | Celestial Dynamics | | Cellular Dynamics | |
|---|---|---|---|---|
| | EPNS | PNS | EPNS | PNS |
| 80 | 9.9 | 7.7 | $-59.6$ | $-326.5$ |
| 400 | 10.2 | 10.0 | $-61.0$ | $-173.9$ |
| 800 | 10.8 | 10.9 | $-59.4$ | $-163.9$ |

**Rollout stability.** As we cannot measure stability by tracking an error metric over time for stochastic dynamics, we resort to domain-specific stability criteria. For celestial dynamics, we define a run to be unstable when the energy in the system jumps by more than 20 units, which we also used to ensure

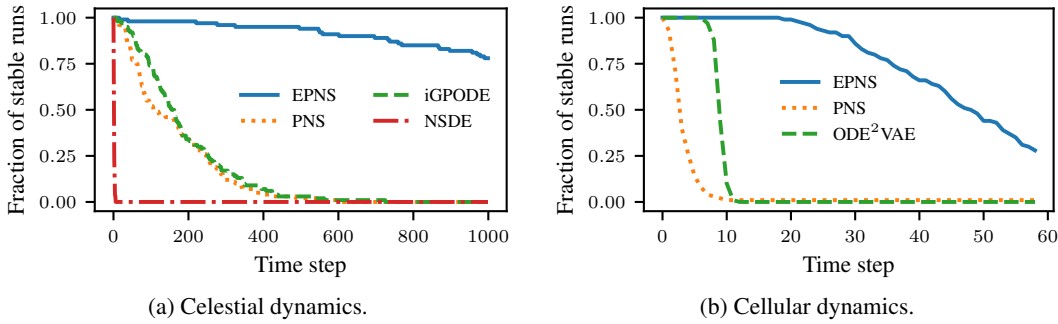

(a) Celestial dynamics.        (b) Cellular dynamics.

Figure 6: Fraction of simulations that remain stable over long rollouts.

the quality of the training samples. For cellular dynamics, we define a run to be unstable as soon as 20% of the cells have a volume outside the range of volumes in the training set. Although these are not the only possible criteria, they provide a reasonable proxy to the overall notion of stability for these applications. The fraction of stable simulations over time, starting from the initial conditions of the test set, is depicted in Figure 6. For both datasets, EPNS generally produces simulations that remain stable for longer than other models, suggesting that equivariance improves rollout stability. Although stability still decreases over time, most of EPNS' simulations remain stable for substantially longer than the rollout lengths seen during training.

# 6 Conclusion

**Conclusions.** In this work, we propose EPNS, a generally applicable framework for equivariant probabilistic spatiotemporal simulation. We evaluate EPNS in the domains of stochastic celestial dynamics and stochastic cellular dynamics. Our results demonstrate that EPNS outperforms existing methods for these applications. Furthermore, we observe that embedding equivariance constraints in EPNS improves data efficiency, stability, and uncertainty estimates. In conclusion, we demonstrate the value of incorporating symmetries in probabilistic neural simulators, and show that EPNS provides the means to achieve this for diverse applications.

**Limitations.** The autoregressive nature of EPNS results in part of the simulations becoming unstable as the rollout length increases. This is a known limitation of autoregressive neural simulation models, and we believe that further advances in this field can transfer to our method [9, 48, 49]. Second, the `SpatialConv-GNN` has relatively large memory requirements, and it would be interesting to improve the architecture to be more memory efficient. Third, we noticed that training can be unstable, either due to exploding loss values or due to getting stuck in bad local minima. Our efforts to mitigate this issue resulted in approximately two out of three training runs converging to a good local optimum, and we believe this could be improved by further tweaking of the training procedure. Finally, our experimental evaluation has shown the benefits of equivariant probabilistic simulation using relatively small scale systems with stylized dynamics. An interesting avenue for future work is to apply EPNS to large-scale systems with more complex interactions that are of significant interest to domain experts in various scientific fields. Examples of such applications could be the simulation of Langevin dynamics or cancer cell migration.

# 7 Acknowledgements

The authors would like to thank Jakub Tomczak and Tim d'Hondt for the insightful discussions and valuable feedback. This work used the Dutch national e-infrastructure with the support of the SURF Cooperative using grant no. EINF-3935 and grant no. EINF-3557.

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

# A  Proof of equivariance

Here, we prove Lemma 1, which we repeat below:

**Lemma 1.** Assume we model $p_\theta(x^{1:T}|x^0)$ as described in Section 3.1, that is, $p_\theta(x^{1:T}|x^0) = \prod_{t=0}^{T-1} p_\theta(x^{t+1}|x^t)$ and $p_\theta(x^{t+1}|x^t) = \int_z p_\theta(x^{t+1}|z, h^t)p_\theta(z|h^t)dz$, where $h^t = f_\theta(x^t)$. Then, $p_\theta(x^{1:T}|x^0)$ is equivariant to linear transformations $\rho(g)$ of a symmetry group $G$ in the sense of definition 2 if:

(a) The forward model is $G$-equivariant: $f_\theta(\rho(g)x) = \rho(g)f_\theta(x)$;

(b) The conditional prior is $G$-invariant or $G$-equivariant: $p_\theta(z|\rho(g)h^t) = p_\theta(z|h^t)$, or $p_\theta(\rho(g)z|\rho(g)h^t) = p_\theta(z|h^t)$;

(c) The decoder is $G$-equivariant with respect to $h^t$ (for an invariant conditional prior), or $G$-equivariant with respect to both $h^t$ and $z$ (for an equivariant conditional prior): $p_\theta(x^{t+1}|z, h^t) = p_\theta(\rho(g)x^{t+1}|z, \rho(g)h^t)$, or $p_\theta(x^{t+1}|z, h^t) = p_\theta(\rho(g)x^{t+1}|\rho(g)z, \rho(g)h^t)$.

To support the proof, we first show that, if a $G$-equivariant distribution $p(y|x)$ exists, i.e. $p(y|x) = p(\rho(g)y|\rho(g)x)$, then the absolute value of the determinant of $\rho(g)$ must equal 1. To this end, let $y_g = \rho(g)y$ and $x_g = \rho(g)x$

$$
\begin{aligned}
1 &= \int_{y_g} p(y_g|x_g)dy_g \\
&= \int_{y_g} p(y|x)dy_g && \{p \text{ is equivariant }\} \\
&= \int_y p(y|x)\,|\det \rho(g)|\,dy && \{\text{substitute } y_g = \rho(g)y\} \\
&= |\det \rho(g)| \int_y p(y|x)dy \\
&= |\det \rho(g)|
\end{aligned}
$$

For the lemma to hold for a first-order Markovian data-generating process, it suffices to have an autoregressive model of which the one-step transition distribution $p_\theta(x^{t+1}|x^t)$ is equivariant. To show this, let $x_g^t = \rho(g)x^t$:

$$
\begin{aligned}
p_\theta(\rho(g)x^{1:T}|\rho(g)x^0) &= p_\theta(x_g^{1:T}|x_g^0) \\
&= \prod_{t=1}^T p_\theta(x_g^t|x_g^{t-1}) && \{\text{Markov property}\} \\
&= \prod_{t=1}^T p_\theta(x^t|x^{t-1}) && \{\text{Equivariant one-step } p_\theta\} \\
&= p_\theta(x^{1:T}|x^0) && \{\text{Inverse Markov property}\}
\end{aligned}
$$

As such, it remains to show that $p_\theta(x^{t+1}|x^t)$ is equivariant when conditions (a)-(c) of the lemma hold. We show here the case where $p_\theta(z|h^t)$ is equivariant, the invariant case is analogous, where $g$ acts on $z$ as $\rho(g)z = Iz$. Again, let $x_g^t = \rho(g)x^t$, and let $z_g = \rho(g)z$:

$$
\begin{aligned}
p_\theta(\rho(g)x^{t+1}|\rho(g)x^t) &= p_\theta(x_g^{t+1}|x_g^t) \\
&= p_\theta(x_g^{t+1}|h_g^t) && \{\text{equivariant } f_\theta\} \\
&= \int_{z_g} p_\theta(x_g^{t+1}|z_g, h_g^t)p_\theta(z_g|h_g^t)dz_g \\
&= \int_{z_g} p_\theta(x^{t+1}|z, h^t)p_\theta(z|h^t)dz_g && \{\text{equivariant } p_\theta\} \\
&= \int_z p_\theta(x^{t+1}|z, h^t)p_\theta(z|h^t)\,|\det \rho(g)|\,dz && \{z_g = \rho(g)z\} \\
&= \int_z p_\theta(x^{t+1}|z, h^t)p_\theta(z|h^t)dz && \{|\det \rho(g)| = 1\} \\
&= p_\theta(x^{t+1}|x^t)
\end{aligned}
$$

## B    Parameters of the Cellular Potts model

Recall that the Hamiltonian for the cell sorting simulation is defined as follows:

$$
H = \sum_{l_i,l_j \in \mathcal{N}(L)} J\left(x(l_i), x(l_j)\right)\left(1 - \delta_{x(l_i),x(l_j)}\right) + \sum_{c \in C} \lambda_V \left(V(c) - V^*(c)\right)^2, \tag{10}
$$

For an explanation on the interpretation of this equation, please consult Section 4.2. The parameter values used in Equation 10 can be found in Tables 4 and 5 below:

Table 4: Parameter values of the Cellular Potts simulation.

| Parameter | Value |
|---|---|
| $J(x(l_i), x(l_j))$ | See Table 5 |
| $\lambda_V$ | 2 |
| $V^*(c)$ | 25 |
| $T$ | 8 |

Table 5: Contact energy $J(x(l_i), x(l_j))$ lookup table.

| Cell type | Medium | Type 1 | Type 2 | Type 3 | Type 4 |
|---|---|---|---|---|---|
| Medium | 0 | 16 | 16 | 16 | 16 |
| Type 1 | 16 | 8 | 11 | 11 | 11 |
| Type 2 | 16 | 11 | 8 | 11 | 11 |
| Type 3 | 16 | 11 | 11 | 8 | 11 |
| Type 4 | 16 | 11 | 11 | 11 | 8 |

These parameters were not chosen to be biologically plausible, but rather to result in behavior in which there is a clear tendency of clustering of same-type cells as time proceeds, and to strike a balance between the stochastic fluctuations and clustering tendencies in the system.

## C    EPNS model details

### C.1    Celestial dynamics

Before going into the specifics, we first discuss some general remarks on the EPNS model implementation. Recall that the data is modeled as a geometric graph consisting of $n$ nodes. The celestial dynamics EPNS model distinguishes between E(n)-invariant features (mass $m_i \in \mathbb{R}^1$) and equivariant features (position $\mathbf{p_i}^t \in \mathbb{R}^3$ and velocity $\mathbf{v_i}^t \in \mathbb{R}^3$) for each node $i$. We provide the Euclidean distance ($||\mathbf{p_i}^t - \mathbf{p_j}^t||_2$), as well as the magnitude of the difference between the velocity vectors ($||\mathbf{v_i}^t - \mathbf{v_j}^t||_2$) between all nodes as edge features $e_{ij}^t \in \mathbb{R}^2$ to the model. ReLU is used throughout the model as activation function. Our implementation of the FA-GNN differs slightly from [38], as we do not construct and invert the frame at each message passing layer. Instead, we construct the frame once, then apply all message passing layers, and then invert the frame again to arrive at invariant or equivariant output. Overall, the model has about 1.8 million trainable parameters.

**Forward model:**

$$\left\{ m \in \mathbb{R}^{n \times 1}, e^t \in \mathbb{R}^{n^2 \times 2}, \mathbf{p^t} \in \mathbb{R}^{n \times 3}, \mathbf{v^t} \in \mathbb{R}^{n \times 3} \right\} \to f_\theta \to$$

$$\left\{ h^t \in \mathbb{R}^{n \times 128}, e^t \in \mathbb{R}^{n^2 \times 2}, \mathbf{p}^t \in \mathbb{R}^{n \times 3}, \mathbf{v}^t \in \mathbb{R}^{n \times 3} \right\}$$

First, the forward model independently embeds each node's mass to an invariant node embedding $h^t \in \mathbb{R}^{128}$, independently of position and velocity attributes. The number of hidden dimensions is kept constant throughout the remainder of the model for the node embeddings. Then, we apply the `FA-GNN` to the graph for 5 message passing layers to update the invariant embedding vectors – the input features are the invariant node embedding as well as the coordinate and velocity vectors of each node. The forward model does not yet update the coordinate and velocity vectors, but simply applies the identity function to the input coordinates and velocities, as we found that it worked better to update these vectors in the decoder only. Note that this trivially implies an equivariant forward model. As such, in this stage, the spatial information is only used to construct a useful E(n)-invariant node embedding.

**Conditional prior:** $\left\{ h^t \in \mathbb{R}^{n \times 128}, e^t \in \mathbb{R}^{n^2 \times 2} \right\} \to p_\theta(z|h^t) \to \left\{ z \in \mathbb{R}^{n \times 16} \right\}$

As we chose a conditional prior that is permutation equivariant, the conditional prior architecture is a message passing GNN with 5 layers that takes as input the $E(n)$-invariant node embeddings produced by the forward model. A linear layer maps the node embeddings to the mean of the conditional prior distribution, while a linear layer followed by Softplus activation maps the embeddings to the scale parameter. We use a 16-dimensional latent space for each node.

**Approximate posterior:**

$$\left\{ h^t \in \mathbb{R}^{n \times 128}, e^t \in \mathbb{R}^{n^2 \times 2}, e^{t+1} \in \mathbb{R}^{n^2 \times 2} \right\} \to q_\phi(z|h^t, x^{t+1}) \to \left\{ z \in \mathbb{R}^{16} \right\}$$

The architecture of the approximate posterior is identical to the conditional prior, with the exception that it accepts additional edge features as input. These are the same E(n)-invariant features that are also given as input to the forward model, but calculated for the system at time $t + 1$, instead of time $t$.

**Decoder:**

$$\left\{ h^t \in \mathbb{R}^{n \times 128}, z \in \mathbb{R}^{16}, e^t \in \mathbb{R}^{n^2 \times 2}, \mathbf{p}^t \in \mathbb{R}^{n \times 3}, \mathbf{v}^t \in \mathbb{R}^{n \times 3} \right\} \to p_\theta(x^{t+1}|h^t, z) \to$$

$$\left\{ \mathbf{p}^{t+1} \in \mathbb{R}^{n \times 3}, \mathbf{v}^{t+1} \in \mathbb{R}^{n \times 3} \right\}$$

The decoder is an `FA-GNN` with three layers. As input, it gets the invariant node embeddings $h^t$ produced by the forward model, the invariant $z$ sampled from the conditional prior (during generation) or approximate posterior (during training), and the position and velocity vectors $\mathbf{p}^t$ and $\mathbf{v}^t$. The decoder equivariantly maps this input to the mean and scale parameters of the Gaussian output distribution. More specifically, the model outputs vectors $\boldsymbol{\mu}^{\Delta p} \in \mathbb{R}^{n \times 3}$ and $\boldsymbol{\mu}^{\Delta v} \in \mathbb{R}^{n \times 3}$. Furthermore, the resulting E(n)-invariant node embeddings are also independently processed by a linear layer with Softplus activation to map to $\boldsymbol{\sigma}^p$ and $\boldsymbol{\sigma}^v$, and by a two layer MLP mapping to a quantity $\Delta_v \in \mathbb{R}$ which is used to scale the velocity's effect on the node position update. Using the quantities described above, the decoder's output distribution is parameterized as follows:

$$\boldsymbol{\mu}^v = \mathbf{v}^t + \boldsymbol{\mu}^{\Delta v}$$
$$\boldsymbol{\mu}^p = \mathbf{p}^t + \Delta_v \cdot \boldsymbol{\mu}^v + \boldsymbol{\mu}^{\Delta p}$$
$$p_\theta(\mathbf{p}^{t+1}|h^t, z) = \mathcal{N}\left(\boldsymbol{\mu}^p, \boldsymbol{\sigma}^p\right)$$
$$p_\theta(\mathbf{v^{t+1}}|h^t, z) = \mathcal{N}\left(\boldsymbol{\mu}^v, \boldsymbol{\sigma}^v\right)$$

The reason for using $\boldsymbol{\mu}^{\Delta p}$ is that we take a relatively large stepsize of $\Delta t = 0.01$, and as such $\boldsymbol{\mu}^v$ might not align with the difference between the coordinates $\mathbf{p}^{t+1} - \mathbf{p}^t$. Accordingly, $\boldsymbol{\mu}^{\Delta p}$ can correct for this difference if necessary.

**Training:** For each epoch, we select one starting point uniformly at random from each trajectory in the training set for the multi-step training. We train the model for 40000 epochs with the Adam optimizer [21] with a learning rate equal to $10^{-4}$ and weight decay of $10^{-4}$. For the KL annealing schedule, we increase the coefficient $\beta$ that weighs the KL term of the loss by 0.005 every 200 epochs. We use a batch size of 64. We do not apply the free bits variant of the ELBO for celestial dynamics.

## C.2 Cellular dynamics

Again, we start with general model remarks before going into the specific components. For the cellular dynamics EPNS model, we get as input a 2D grid of dimensions $h \times w$ with two channels: the first channel contains integer values between 0 and $c$, indicating the index of each cell, where 0 stands for background and $c = |C|$ is the number of cells. The second channel contains integer values between 0 and $\tau$ that indicate the cell type, where 0 is again background, and $\tau \leq c$ is the number of different cell types. We again use ReLU activation throughout the model as nonlinearity. Overall, the model has around 13 million trainable parameters.

We first describe the overall model structure in terms of the forward model, conditional prior, and decoder, after which we describe the `SpatialConv-GNN` architecture for this model in more detail.

**Forward model:** $\{x^t \in \mathbb{Z}^{h \times w \times 2}\} \to f_\theta \to \{h^t \in \mathbb{R}^{c \times h \times w \times 32}\}$

The forward model first one-hot encodes both the index and type channels of the input $x^t$, yielding two grids of shape $h \times w \times c$ and $h \times w \times \tau$. Then, each of the channels in the one-hot encoded cell index grid is seen as a separate node, and the one-hot encoded type channel tensor is concatenated to each node along the channel axis. Note that we do not assume permutation equivariance of the cell types, but only of the cell indices. This means we now have $c$ grids of shape $h \times w \times (\tau + 1)$, which we simply store as one grid of shape $c \times h \times w \times (\tau + 1)$. Subsequently, a linear convolution embeds the grid to an embedding dimension of 32, resulting in a tensor of shape $c \times h \times w \times 32$. Then, a `SpatialConv-GNN` with a single message passing layer processes this tensor, again resulting in a grid of shape $c \times h \times w \times 32$.

**Conditional prior:** $\{h^t \in \mathbb{R}^{c \times h \times w \times 32}\} \to p_\theta(z|h^t) \to \{z \in \mathbb{R}^{c \times 64}\}$

The conditional prior independently maps each cell to a latent variable $z$ by applying a convolutional layer with ReLU activation, followed by three repetitions of $\{\text{convolution} \to \text{ReLU} \to \text{MaxPool}\}$. All convolution layers have a kernel size of $9 \times 9$, and we perform $2 \times 2$ maxpooling. Then, global meanpooling is performed over the spatial dimensions. Subsequently, a linear layer followed by ReLU activation is applied. The resulting vector is mapped by a linear layer to the mean of the conditional prior distribution, while a linear layer followed by Softplus activation maps the same vector to the standard deviation of the conditional prior.

**Approximate posterior:**

$$\{h^t \in \mathbb{R}^{c \times h \times w \times 32}, x^{t+1} \in \mathbb{Z}^{h \times w \times 2}\} \to q_\phi(z|h^t, x^{t+1}) \to \{z \in \mathbb{R}^{c \times 64}\}$$

The architecture of the approximate posterior is almost identical to the conditional prior. The only exception is that first, $x^{t+1}$ is encoded to nodes in the same way as described in the forward model, resulting in a binary tensor $x^{t+1} \in \{0, 1\}^{c \times h \times w \times (\tau+1)}$. For each cell, this tensor is concatenated to the input along the channel axis, and is then processed by a model with the same architecture as the conditional prior.

**Decoder:**

$$\{h^t \in \mathbb{R}^{c \times h \times w \times 32}, z \in \mathbb{R}^{c \times 64}\} \to p_\theta(x^{t+1}|h^t, z) \to \{x^{t+1} \in \mathbb{Z}^{h \times w \times 2}\}$$

For each cell, the sampled latent variable $z$ is concatenated along the channel axis for all pixels in the grid. The decoder then consists of a `SpatialConv-GNN` with a single message passing layer, followed by a linear convolutional layer with kernel size $1 \times 1$ that maps to a single channel. The resulting tensor has shape $c \times h \times w \times 1$. We then apply Softmax activation along the cell axis to get the pixel-wise probabilities over cell indices. We only optimize the likelihood for the cell index, not for the cell type, as the cell type can be determined by the cell index. Specifically, to get a

(discretized) output where the cell type is included, we simply take the cell index with the highest probability and match it with the corresponding cell type, which we can extract from the known input $x^t$.

SpatialConv-GNN: Recall the SpatialConv-GNN message passing layer from Equation 9:

$$h_i^{l+1} = \psi^l \left( h_i^l, \bigoplus_{j \in \mathcal{N}(i)} \phi^l(h_j^l) \right).$$

The core components that define the SpatialConv-GNN are $\phi$, $\bigoplus$, the selection of the neighbors $\mathcal{N}(i)$, and $\psi$. We describe the details of each of the components below:

- $\phi$: First, $\phi$ performs a strided convolution with kernel size $2 \times 2$ and a stride of 2 to downsample each grid $h_j$. Then, three repetitions of {convolution $\rightarrow$ ReLU} are applied, where the convolutions have a kernel size of $9 \times 9$.

- $\bigoplus$: We use elementwise mean as permutation-invariant aggregation function.

- $\mathcal{N}(i)$: We opted for a fully-connected graph, meaning $\mathcal{N}(i) = C$ for all $i \in C$. The reasons for this are twofold: on the one hand, the convolutional nature of $\phi$ and $\psi$ already takes spatial locality into account. On the other hand, using all cells as neighbors for a single cell $i$, including $i$ itself, means that we only have to compute $\bigoplus_{j \in \mathcal{N}(i)} \phi^l(h_j^l)$ once and can re-use it for all $i \in C$, substantially reducing memory requirements.

- $\psi$: $\psi$ first upsamples the aggregated messages using a transposed convolution to the original grid size. Then, the result is concatenated along the channel axis of $h_i$. Subsequently we apply a simple UNet, using three downsampling and upsampling blocks. The convolutional layers in these blocks have a kernel size of $5 \times 5$. We also tried to parameterize $\psi$ with only ordinary convolutional layers, but found that the UNet's ability to capture dynamics at different scales substantially helped performance. However, the aliasing effects introduced from the downsampling and upsampling might result in translation equivariance not actually being achieved. Nevertheless, permutation equivariance is the most crucial symmetry to respect in this setting. Furthermore, the architecture of $\psi$ remains convolutional, and consequently it still enjoys the benefits of inductive biases related to locality and approximate translation equivariance.

**Training:** For each epoch, we select one random starting point uniformly at random from each trajectory in the training set for the multi-step training. We train the model for 180 epochs with the Adam optimizer [21] with a learning rate equal to $10^{-4}$, a weight decay of $10^{-4}$, $\beta_1 = \beta_2 = 0.9$, and $\varepsilon = 10^{-6}$. For the KL annealing schedule, we increase the coefficient $\beta$ that weighs the KL term of the loss by 0.04 every epoch, until a maximum of $\beta = 1$. Furthermore, we use the free bits modification of the ELBO to prevent posterior collapse [22]:

$$\text{ELBO}_{\text{free bits}} =$$

$$\mathbb{E}_{q_\phi(z|x^t,x^{t+1})} \left[ p_\theta(x^{t+1}|z,x^t) \right] - \beta \sum_{j=1}^{|z|} \text{maximum} \left( \lambda, KL \left[ q_\phi(z_j|x^t,x^{t+1}) || p_\theta(z_j|x^t) \right] \right),$$

with $\lambda = 0.1175$. In the above objective function, the KL terms for each $z_j$ are summed over the cells and averaged over a minibatch. We use a batch size of 8.

## D   Baseline model details

### D.1   Celestial dynamics

**PNS**   The PNS model for celestial dynamics is identical to the EPNS model described in Appendix C.1, with a few exceptions. Instead of an equivariant FA-GNN architecture for the forward

model and decoder, we use an ordinary message passing GNN. Consequently, we do not distinguish between E(n)-equivariant and invariant features anymore, but simply concatenate all node features to one node feature vector as input. The training procedure is identical to the EPNS model. Similar to the EPNS model, this model has approximately 1.8 million trainable parameters.

**iGPODE** For the iGPODE baseline, we used the code from the original paper [59].[2] In terms of hyperparameter optimization, we consider combinations of: $\{100, 500, 2000\}$ inducing points for the sparse Gaussian process; whether or not to use an initial value encoder; and Softplus versus ReLU activations. We selected the model with the best validation loss, which had 100 inducing points, an initial value encoder, and ReLU activations.

**NSDE** We train the NSDE as a latent SDE, like [27] but without the adjoint sensitivity method. The model consists six networks: a context encoder, f-net, h-net, g-net, initial encoder, and a decoder. The context encoder consists of a bidirectional GRU, and encodes the data into a context vector at each time step to be used by the drift of the encoding distribution. The f-net parameterizes the drift of the encoding distribution and the h-net that of the prior distribution. The g-net parameterizes the diffusion of both the encoding and prior distribution. For this we tried both the 'diagonal' and 'general' noise options in TorchSDE [20, 27], but settled on 'diagonal' noise as it performed significantly better. Contrary to [27], the initial encoder does not depend on the context vector at the initial time step because we did not want it to suffer from the GRU performing poorly on sequences of one time step in the sampling experiments. Instead, it uses a fully connected network with skip connections to process the data at the first time step. Because the data comes from a Markov process and the full state of the system is available to the network, this should suffice to parameterize the initial distribution. The prior against which this is measured is a learned mixture of multivariate Gaussians to allow the model a large degree of flexibility for the initial distribution. This prior plays no role in the evaluation of the model, however.

For the decoding distribution, we experimented with both an autoregressive distribution and with a non-autoregressive distribution. The non-autoregressive distribution worked significantly better than the autoregressive one. In both cases, we use a fully connected neural network with skip connections to parameterize a Gaussian distribution with diagonal covariance. We also experimented with parameterizing the covariance and keeping it fixed. The fixed value for the variance worked better.

The initial encoder and prior have their own, higher, learning rate separately from the parameters of the other networks. Both sets of parameters were optimized using Adam [21]. During hyperparameter optimization we found that a learning rate of $0.001$ for the initial part of the model and one of $0.0005$ for the other networks, combined with exponential learning rate scheduling with $\gamma = 0.9999$ for both sets of parameters resulted in consistent and stable training. Moreover, we used weight decay with value $0.0001$. The latent size and context size we settled on are $64$ and $256$ respectively. The hidden size of the f-, g-, and h-net are $128$ (with two hidden layers), and that of the decoding distribution is $140$ with three hidden layers. Finally, we restricted the length of the sequences on which we trained to 20 and used batches of size 256.

### D.2 Cellular Dynamics

**PNS** As non-equivariant baseline, PNS models the cellular dynamics using a modern UNet backbone, which has shown state-of-the-art performance in modeling spatiotemporal dynamics on a grid [14]. Notably, the PNS model for cellular dynamics has about an order of magnitude more trainable parameters than its EPNS counterpart: around 127 million parameters, as opposed to 13 million for EPNS. We use ReLU activations and a hidden dimension of 128 throughout the model.

First, the input is one-hot encoded the same way as done for EPNS (see Appendix C.2). Then, it is processed by the model components which are defined as follows:

- **Forward model:** a modern UNet architecture with wide ResNet blocks, spatial attention, downsampling layers, and group normalization, as described in [14]. The architecture consists of three downsampling and upsampling steps, with attention only being applied in

---

[2]Available at https://github.com/boschresearch/igpode at the time of writing.

the lowest layer of the UNet. The number of channels is 128 at the top of the UNet, and doubled after each spatial downsampling step to a maximum of 512 channels.

- **Conditional prior:** the conditional prior architecture is identical to the one used in EPNS, except that the number of channels used in the convolutional layers is 128 as opposed to 32.

- **Approximate posterior:** The approximate posterior architecture is the same as the conditional prior, with the exception that it accepts the one-hot encoding of $x^{t+1}$ as additional input channels.

- **Decoder:** the decoder consists of 4 convolutional layers with kernel size $3 \times 3$, followed by Softmax activation at the end to map to pixel-wise probabilities over the cell indices.

- **Training procedure:** as for EPNS, we train PNS with multi-step training for 180 epochs, with the same KL annealing schedule. We use a free bits parameter of $\lambda = 0.1$ to prevent posterior collapse. For optimization, we use Adam with a learning rate of $10^{-4}$ and a weight decay of $10^{-4}$.

**ODE$^2$VAE**    Our ODE$^2$VAE implementation is adapted from the PyTorch implementation of [58].[3] Specifically, we changed the expected shapes of the convolutional layers to match with the shape of the cellular dynamics data, and changed the decoding distribution to a categorical distribution instead of a Bernoulli distribution. The hyperparameters we search over are: $\{4, 5\}$ convolutional layers in the encoder and decoder; $\{16, 32\}$ filters in the initial encoder layer and last decoder layer (which are doubled in each subsequent convolutional layer); and $\{32, 64\}$ dimensions for the latent space. We disregarded model configurations that did not fit within the 40GB of GPU VRAM with a batch size of 8 to keep the computational requirements reasonable. The best-performing model in terms of validation loss has a 64 dimensional latent space, 32 filters in the initial convolutional layer, and 4 convolutional layers in the encoder and decoder, which has around 21 million trainable parameters.

## E    Additional results

### E.1    Ablation studies

We investigate the following ablations:

- Changing the design choice of using permutation invariant or permutation equivariant latent variables, giving insight into the importance of this decision;

- A deterministic variant of the EPNS framework, which does not have a latent space but autoregressively produces point predictions. This gives insight into the need of a probabilistic model in the stochastic setting;

- An ablation of the EPNS framework which does not have a latent space but rather produces $p_\theta(x^{t+1}|x^t)$ directly, giving insight into the need of the latent variable approach. As opposed to the main model, where we sample $z$ but construct $x^{t+1}$ such that $p_\theta(x^{t+1}|z, h^t)$ is maximized as is commonly done in VAEs, here we sample from the decoder distribution $p(x^{t+1}|h^t)$, which does not depend on $z$;

- A non-equivariant PNS model with MLP backbone architecture, investigating the need for permutation equivariance in n-body dynamics.

Since the multi-step training heuristic depends on steering the trajectory using the latent space, the deterministic and no-latent ablations were trained with one-step training. All other parameters and methods were kept constant.

Overall, Table 6 shows that the main EPNS configuration with equivariant latent variables either outperforms or performs competitively with its ablations, in terms of LL and $D_{\text{KS}}$. More specifically, a few observations stand out:

- The EPNS variant without latent variables outperforms EPNS in terms of LL for celestial dynamics. This can be explained by the following two reasons: first, EPNS-no latents is trained using single step training, which purely optimizes the one-step transition probability.

---

[3]Available at https://github.com/cagatayyildiz/ODE2VAE at the time of writing.

Table 6: Results for variants of the EPNS framework on the celestial dynamics application.

| | PNS | EPNS | EPNS-invariant latents | EPNS-no latents | EPNS-deterministic | PNS-MLP |
|---|---|---|---|---|---|---|
| ↑LL ($\cdot 10^3$) | 10.9±0.4 | 10.8±0.1 | 11.6±0.1 | **13.2±0.1** | N/A | 5.9±0.3 |
| ↓$D_{\text{KS}}(KE)$ t=50 | 0.61±0.18 | **0.14**±0.03 | 0.39±0.12 | 0.25±0.12 | 0.63±0.08 | 0.52±0.21 |
| ↓$D_{\text{KS}}(KE)$ t=100 | 0.20±0.06 | **0.14**±0.04 | 0.18±0.02 | 0.19±0.05 | 0.7±0.05 | 0.44±0.11 |

Table 7: Results for variants of the EPNS framework on the cellular dynamics application.

| | PNS | EPNS | EPNS-invariant latents | EPNS-no latents | EPNS-deterministic |
|---|---|---|---|---|---|
| ↑LL ($\cdot 10^4$) | -16.4±0.3 | **-5.9±0.1** | -27.5±15.4 | -9.9±0.0 | N/A |
| ↓$D_{\text{KS}}$(#clusters) t=30 | 0.70±0.10 | **0.58±0.09** | **0.60±0.22** | 1.00±0.00 | 0.93±0.08 |
| ↓$D_{\text{KS}}$(#clusters) t=45 | 0.77±0.05 | **0.58±0.05** | **0.55±0.25** | 1.00±0.00 | 0.97±0.03 |

Since the log-likelihood is calculated as the sum of one-step transition probabilities for an autoregressive model, this is perfectly aligned with the LL metric. On the other hand, EPNS is trained with multi-step training, which enables it to produce better rollouts – see also the $D_{\text{KS}}$ values in Table 6 – but at the cost of not directly optimizing for LL. The second reason is that the step size of $\Delta t = 0.1$ is relatively small. This means that the assumption of a Gaussian one-step distribution made by EPNS-no latents is reasonable.

- In terms of $D_{\text{KS}}$, EPNS with global, permutation-invariant latent variables performs competitively with EPNS when taking the average over training runs, as seen in Table 7. However, the variance is much higher. Consequently, training EPNS with invariant latents is much more unstable than EPNS with equivariant latents. Moreover, the LL metric is always better for EPNS with equivariant latents in this setting.

Further, we also investigate the rollout stability of the various ablations – see Section 5.2 for the details on how this is calculated. For the celestial dynamics problem, Figure 7a shows that the main EPNS configuration with permutation equivariant latents tends to produce simulations that remain stable for slightly longer than its counterpart with invariant latents. Moreover, both the permutation equivariant and permutation invariant alternatives of EPNS remain stable for substantially longer than variants without any latent variables, or a deterministic variant. Further, Figure 7b shows that PNS with an MLP backbone has poor rollout stability, which is in line with its poor performance across the other metrics.

For the cellular dynamics problem, as shown in Figure 8, the deterministic variant of EPNS appears to produce more stable simulations at first glance. However, upon further inspection, this is only the case because the deterministic model tends to produce 'frozen' dynamics, where cells move at much slower speeds than expected. Specifically, the average velocities with which cells move, expressed in grid sites per time step, are 1.82 for the ground-truth dynamics, 1.35 for EPNS, and 0.13 for the deterministic ablation of EPNS. As such, the high stability values for EPNS-deterministic shown in Figure 8 can be considered an artifact of the stability metric, rather than the result of truly stable and realistic simulations. In fact, a model that simply propagates the initial state $x^0$ would retain 100% stability indefinitely. Consequently, EPNS with permutation equivariant latents produces the most stable simulations where cells move at reasonable speeds, albeit still slower than in the ground-truth.

### E.2 Phase space coverage for celestial dynamics

We also investigated the phase space that is explored by the ground truth and EPNS, PNS and PNS with an MLP backbone respectively. To this end, we generate histograms of the values of the positions and velocities taken by the object with index 1, both for the ground truth and the model. Note that these histograms show empirical densities that have been aggregated over time and over 100 runs all starting from the same initial condition, with different random seeds. The results are shown in Figure 9 (EPNS), Figure 10 (PNS) and Figure 11 (PNS-MLP). Visually, the phase space explored by

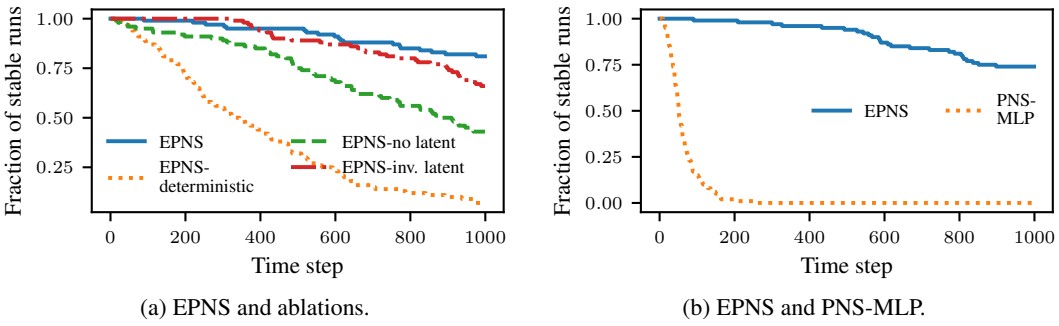

(a) EPNS and ablations.

(b) EPNS and PNS-MLP.

Figure 7: Fraction of simulations that remain stable over long rollouts for celestial dynamics.

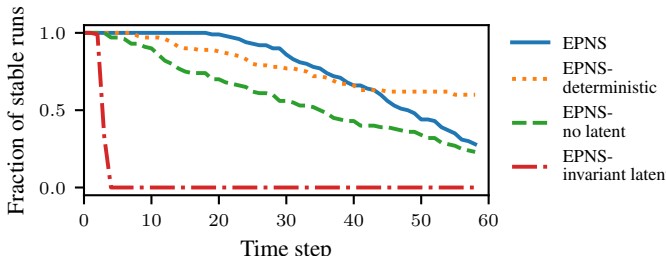

Figure 8: Fraction of simulations that remain stable over long rollouts for cellular dynamics.

both EPNS and PNS appears to overlap well with the ground truth, and their performance appears similar. In contrast, the coordinates and velocities explored by PNS-MLP do not overlap well with the ground truth. This suggests that the permutation symmetry, respected by both PNS and EPNS, is vital in designing a good model for these systems, even when they consist of only five bodies.

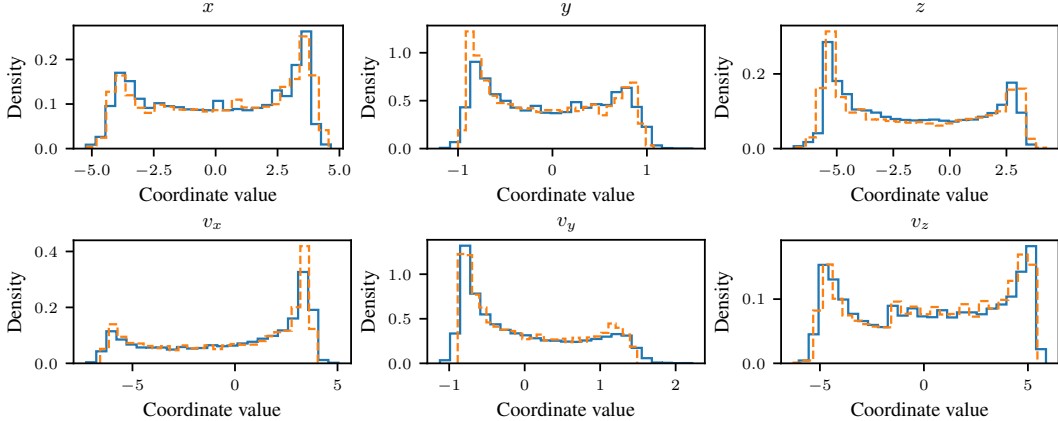

Figure 9: Phase-space coverage plots of the n-body system for EPNS. The plots show histograms of phase-space coordinates of the body with index 1, aggregated over time and 100 samples starting from the same initial condition. The solid blue line indicates the histograms obtained from the ground-truth simulator, while the dashed orange line indicates the histograms obtained from the model.

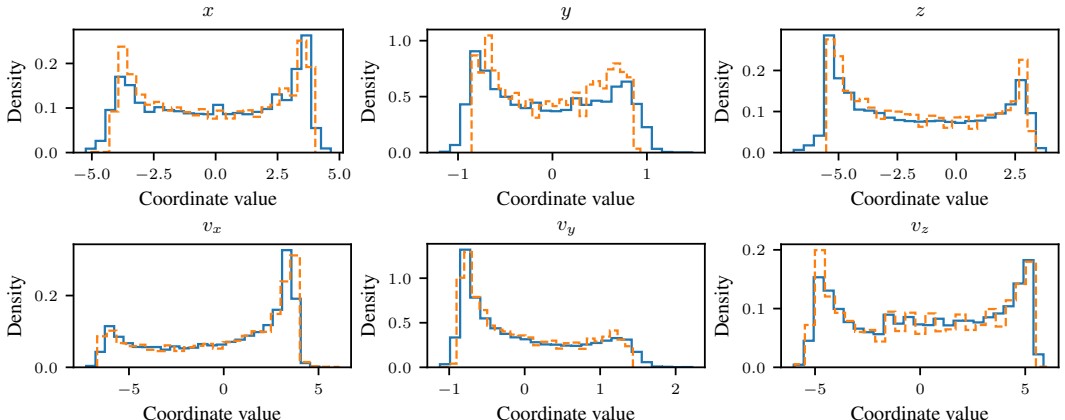

Figure 10: Phase-space coverage plots of the n-body system similar to Figure 9, but with PNS as model.

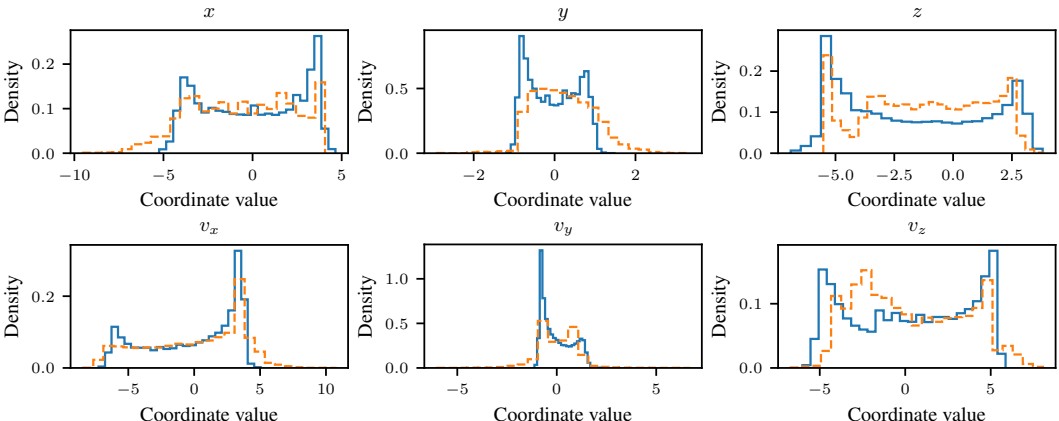

Figure 11: Phase-space coverage plots of the n-body system similar to Figure 9, but with PNS with MLP backbone as model.

### E.3 20-body system results

In this section, we report results of applying EPNS to a 20-body system, as opposed to a 5-body system in Section 5. Although this system is not as high-dimensional as most challenging potential real-life applications, the 20-body system can help to assess the potential of scaling EPNS to such systems. Notably, for these experiments, we do not perform any hyperparameter optimization relative to the 5-body EPNS model.

Table 8 shows the quantitative results of applying EPNS and PNS to a 20 body system. Similar to the 5-body system, we see that EPNS outperforms PNS on all metrics. Note that LL is calculated by summing over all dimensions, and since we have 20 bodies here, LL is higher than the 5-body model. If we were to calculate the per-body LL instead, it would be lower, as expected for a more challenging problem setup without any specific hyperparameter optimization. Surprisingly, the $D_{\text{KS}}$ scores of EPNS are similar to those of the 5-body problem, indicating that EPNS has the potential to generalize to larger systems.

Further, Figure 12 shows the distribution of the kinetic energy over time, both of the ground truth and of EPNS and PNS, calculated over 100 simulations starting from the same initial conditions, similar to Figure 5. Again, we observe that the distribution generated by EPNS overlaps very well with the ground-truth, and notably better than the distribution generated by PNS.

Finally, similar to Section E.2, we plot the empirical density of the phase space coordinates of the body with index 1, aggregated over time and over 100 samples starting from the same initial condition but with different random seeds. Again, the density produced by EPNS is plotted in dashed orange lines, while the ground truth is plotted in solid blue. Similar to Figure 9, we observe that both densities overlap very well, suggesting that the distribution over trajectories from a fixed initial condition produced by EPNS matches the ground truth, also in this higher-dimensional setting.

Table 8: Results for scaling EPNS to a 20-body system.

|  | PNS | EPNS |
| --- | --- | --- |
| $\uparrow$ LL ($\cdot 10^3$) | 15.4$\pm$0.1 | **15.6**$\pm$0.0 |
| $\downarrow D_{\text{KS}}$(KE) t=30 | 0.22$\pm$0.03 | **0.14**$\pm$0.03 |
| $\downarrow D_{\text{KS}}$(KE) t=45 | 0.21$\pm$0.03 | **0.14**$\pm$0.05 |

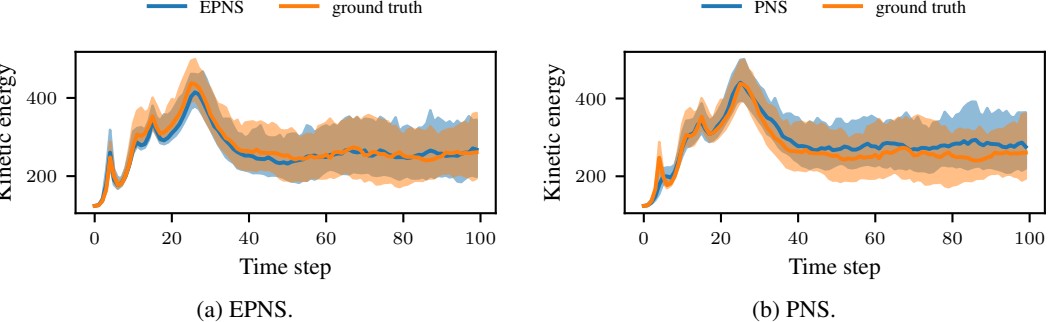

(a) EPNS.  (b) PNS.

Figure 12: Ground-truth distribution over kinetic energy for a 20-body system compared to EPNS and PNS.

### E.4 Additional samples for celestial dynamics

We provide additional qualitative results for celestial dynamics in Figure 14. To do so, we select four samples from the test set uniformly at random. Overall, we observe that EPNS, iGPODE and PNS generally produce simulations that look visually plausible. NSDE does not generate plausible new simulations. We attribute this primarily to the relatively high KL divergence component (around 1900) of the NSDE model ELBO values (shown in Table 1). Recall that, following the same procedure as for the other baselines, we selected the NSDE hyperparameters that resulted in the best validation ELBO value, which might not have been optimal in terms of new sample quality. Notably, all models

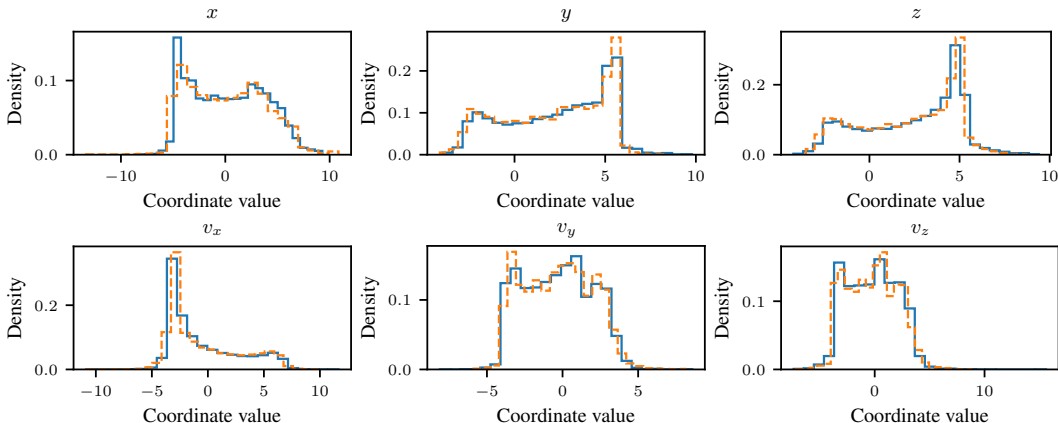

Figure 13: Phase-space coverage plots of the n-body system similar to Figure 9, but with EPNS trained on a celestial dynamics with 20 bodies.

can struggle with handling close encounters, and a body may get 'stuck' to another body, for example shown by the red body in the bottom right trajectory. Generally, EPNS seemed to struggle less with handling such close encounters, although the top left trajectory suggests that such behavior is not completely eliminated from the model.

### E.5 Additional samples for cellular dynamics

Additional qualitative results for cellular dynamics are shown in Figure 15. Again, we selected four samples from the test set uniformly at random. EPNS produces samples that look realistic, as the cells generally have reasonable sizes, show dynamic movement and cell shapes, and tend to cluster together with other cells of the same type over time. Still, sometimes the volumes of some cells become unreasonably small after many rollout steps, which is the main cause for its stability decreasing, as shown in Figure 6b. In the case of PNS, although it does exhibit clustering behavior, cells often quickly evolve to have unrealistic volumes, also reflected in the rapidly decaying stability shown in Figure 6b. ODE$^2$VAE generally produces simulations in which cells look 'frozen' and lack clustering behavior. We attribute this to the fact that ODE$^2$VAE models the dynamics in latent space, and struggles to propagate the geometric structure of the data as time proceeds as a result.

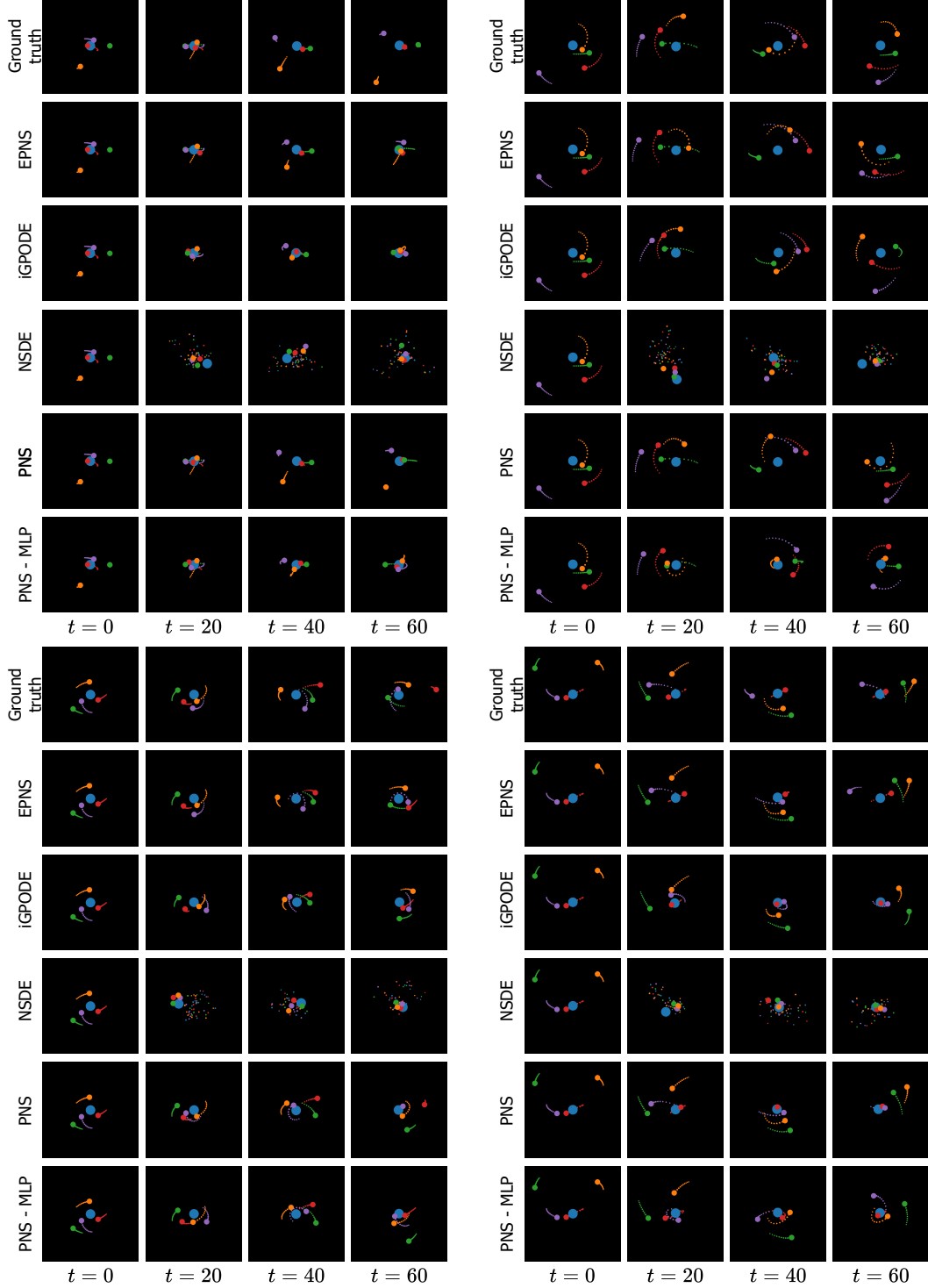

Figure 14: Additional qualitative results for celestial dynamics simulation.

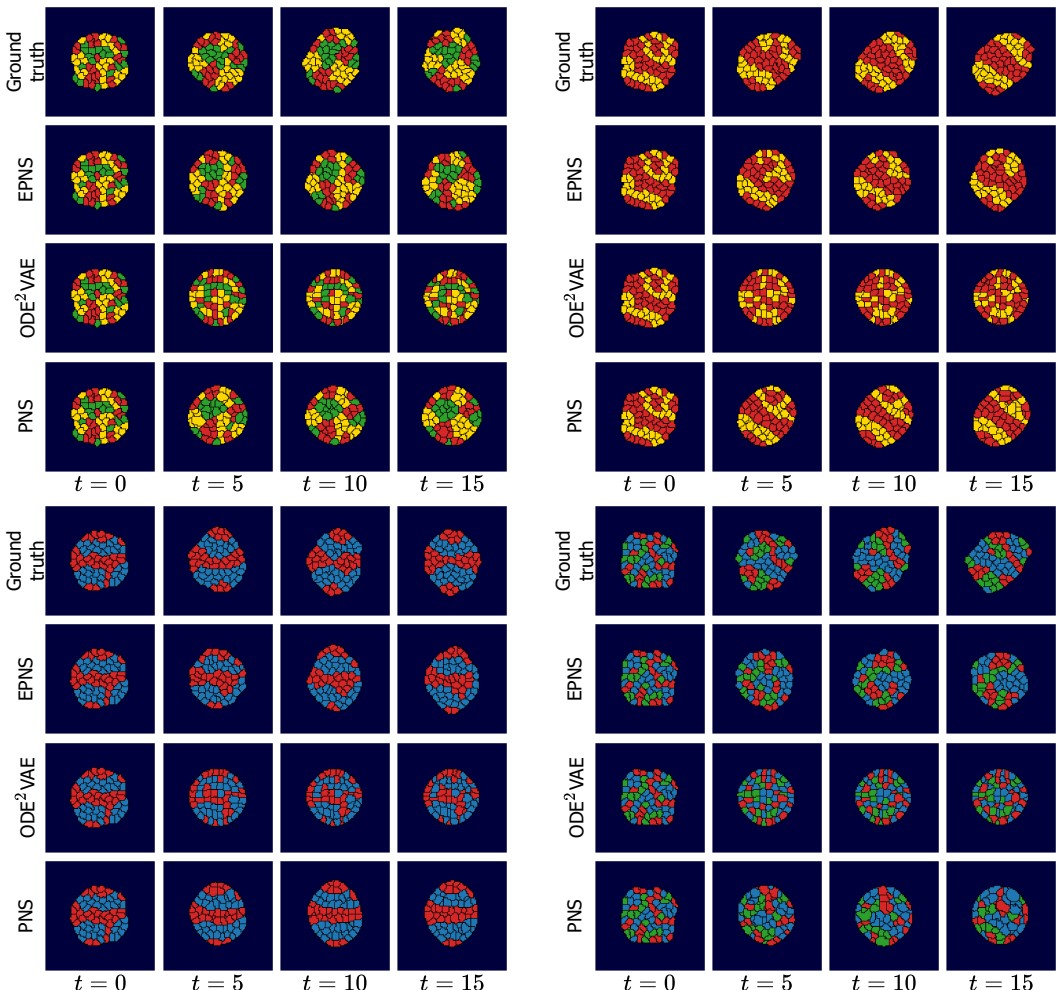

Figure 15: Additional qualitative results for cellular dynamics simulation.

