# OpenReview forum: "Equivariant Neural Simulators for Stochastic Spatiotemporal Dynamics"
_NeurIPS.cc/2023/Conference — NeurIPS 2023 poster_

### Official Review · Reviewer_yRZx · 2023-07-06

**Soundness:** 3 good
**Presentation:** 3 good
**Contribution:** 2 fair
**Rating:** 5
**Confidence:** 4

**Summary:**

This paper proposed to incorporate symmetries into probabilistic models to simulate stochastic dynamics over trajectories. In particular, the authors developed EPNS, an equivariant version of sequential CDVAE for the generation of the state transition. The results on two tasks including stochastic n-body system and stochastic cellular dynamics, show that EPNS considerably outperforms existing neural network-based methods for probabilistic simulation in terms of simulation quality, data efficiency, rollout stability, and uncertainty quantification.

**Strengths:**


1.	It is valuable to design equivariant probabilistic neural simulators against existing deterministic counterparts. It is true that random effects exist in common systems and should be characterized.

2.	Lemma 1 provides convincing derivations for the formulation of equivariant CDVAE. It is interesting to see that there are two different ways to derive equivariant models: invariant latent and equivariant latent. Besides, severe deviation over long rollout simulations has always been a tricky problem. To mitigate this phenomenon, the paper adopts a heuristic method by using the ground truth of the next state to steer the sampling progress， which is inspiring.

3.	This paper is well-organized, first introducing the general formulation and then detailing the specific architectures for two interesting applications. I enjoy the writing. The experimental evaluations, although insufficient, is appropriately demonstrated to support certain claims of the authors.


**Weaknesses:**


1.	My biggest concern is that one important reference [A] is not cited and discussed. The authors claim that “current probabilistic methods do not incorporate general domain symmetries into their model architectures”, which is incorrect if checking [A]. The work by [A] has developed equivariant Gaussian Process (GP) to model stochastic fields, including scalar, vector and stacked fields. Equivariant GP is applicable to the task of modeling system evolutions discussed in this paper. We can directly consider p(x_{t+1}|x_t) as the prediction of vector fields studied in [A]. More importantly, when referring to [A], many claims of the authors in Introduction (Lines 47-51, Lines 55-57) and related works (Line 95-97) are no longer correct, and [A] should be at least considered as an indispensable baseline in the experiments. It is unclear how much novelty remains and how many extra efforts should be made to reflect the modification of discussing [A].

2.	For the experimental evaluations, several crucial contribution points are NOT justified sufficiently. First of all, the equivariant deterministic methods should be tested, by which we can observe why we need stochastic modeling. Second, besides the reference [A] mentioned above, the authors are suggested to compare against typical probabilistic models (such as ODEVAE) with trivial equivariant modification, which is able to better support the benefit of the proposed designed model. Third, in comparison with using the latent model z, one can directly model the transition probability p(x_{t+1}|x_t) straightforwardly via multiple equivariant layers. How does this baseline perform?

3.	The authors proposed to use invariant latent for Celestial dynamics, and equivariant latent for cellular dynamics. Certain ablations are suggested to provide to support this design.

4.	The conclusion of Lemma 1 is interesting. It provides sufficient conditions to maintain the symmetries, but are these conditions also necessary?

5.	Would you please explain more on why EGNN is unstable for the Celestial dynamics task?

6.	Is it possible to combine invariant latent and equivariant latent into one single model. In this case, does it mean that invariant latent corresponds to invariant features in EGNN and equivariant latent corresponds to equivariant features in EGNN?

7.	How to generate stochastic ground truth trajectories in the experiments? How to compute quantile interval of the observations in Figure 5.


[A] Equivariant Learning of Stochastic Fields: Gaussian Processes and Steerable Conditional Neural Processes, ICML 2021.


**Questions:**

See the weakness part above.

**Limitations:**

The authors have adequately discussed the limitations.

---

> ### Author Rebuttal · Authors · 2023-08-09
>
> Thank you for your thorough review. We appreciate that you find equivariant probabilistic neural simulation a valuable contribution, that you find our methodology convincing and inspiring, and that you liked the presentation of the paper. Please find our responses to the your specific comments below:
>
> **W1:** Thank you for bringing this paper to our attention. We agree that [A] provides a method for equivariant modeling stochastic phenomena, and we will update our related work discussion accordingly. However, the main method described in [A], SteerCNPs, is not a natural fit to the problem setting considered in our work, as it is a method for spatial interpolation rather than temporal dynamics simulation. More specifically, as described in 5.3 of [A], it requires a context set Z to encode, which in their experiments are, for example, certain pixel values of MNIST data that have been observed, or wind speeds that have been recorded only at certain locations. In our setting, such a context set is not available. Additionally, from Section 5.1 in [A] it is clear that the SteerCNP method is exponential in n, which makes it infeasible for the problems we consider: for the 5-body system, n would be 30. In fact, [A] poses the use of point-cloud based techniques to overcome this limitation for scenarios like ours as an interesting avenue for future work.
>
> To directly model the transition probability as a neural field distributed according to a Gaussian Processes, as you suggest, would be more akin to learning an equivariant GP prior, rather than the posterior [A] focusses on. Although [A] provides sufficient and necessary conditions for a GP prior to be invariant (theorem 1), these conditions need modification for the GP to be equivariant. Moreover, [A] does not provide methods for parameterizing the kernel and mean of a GP prior in a way that it is learnable, flexible, and satisfies the relevant conditions for in- or equivariance given a transformation group G.
>
> However, we see that without context the specific lines in our paper that you pointed out appear as if we claim to be the first paper to model equivariant stochastic phenomena in general. Besides updating the related work section to discuss this work, we will also update these text segments to stress that our claims concern modeling dynamical systems using equivariant distributions over trajectories with regard to general symmetry groups specifically.
>
> **W2 & W3:** Thank you for suggesting these additional experimental evaluations. We have now run additional experiments with the following ablations: a deterministic variant of EPNS, a variant that models the transition probability directly, and an ablation on using in-/equivariant latents. The main results can be found in Tables R1 and R2 and Figures 1a, 1c and 3b in the general response. For celestial dynamics, it turned out that a model with permutation equivariant latents worked even better than our original configuration with invariant latents in terms of stability and the D_KS scores. For cellular dynamics, the original configuration with equivariant latents substantially outperforms the other variants of the model.
>
> As for comparing against modifications of probabilistic models, the architectural adaptations required to make the baseline models equivariant to the relevant transformation group are not trivial. For example, for the n-body system, introducing E(3)-equivariance in the iGPODE and Neural SDE models would require a complete overhaul of the model structure. For the cellular dynamics, introducing permutation and translation equivariance in ODE2VAE would require the integration of our SpatialConv-GNN, which is practically infeasible since both already have high memory requirements by themselves. Of course, the question of how to build equivariant architectures into various generative models is an interesting avenue of research. Our paper provides a promising step in this direction.
>
> **W4:** The conditions in this lemma are sufficient but not necessary: for example, if the decoder ignores the latent state entirely, the entire model can be G-equivariant even if the conditional prior is neither invariant nor equivariant. However, doing so considerably limits the class of distributions that can be learned.
>
> **W5:** We did preliminary experiments with EGNN for the architecture, but the multi-step training would often become unstable with the EGNN. As such, we switched to the FA-GNN for practical reasons, as instability issues seemed to occur less frequently. Furthermore, the FA-GNN has also been shown to outperform the EGNN for n-body dynamics prediction (see Section 5.3 of [B]), so the choice for the FA-GNN is also motivated from a performance perspective.
>
> **W6:** It is possible to combine both invariant and equivariant latent variables within the EPNS model. Whether doing so effectively separates invariant and equivariant features seems plausible, but further research is necessary to confirm if this is the case.
>
> **W7:** We will update the appendix with the equations and algorithms that are used for simulating the ground truth trajectories. Note that the code for generating the ground-truth dynamics can be found in the supplemental material. The quantile intervals in Figure 5 have been computed by running the ground-truth simulator and the ML model 100 times from the same initial state, and measuring the potential energy (resp. number of cell clusters) of the system at each point in time. We then calculate the (0.1, 0.9) quantiles of these values at each time point.
>
> [A] P. Holderrieth et al. “Equivariant Learning of Stochastic Fields: Gaussian Processes and Steerable Conditional Neural Processes”, ICML 2021.
>
> [B] O. Puny et al. “Frame Averaging for Invariant and Equivariant Network Design”, ICLR 2022

---

> > ### Comment · Reviewer_yRZx · 2023-08-18
> > **Thank you**
> >
> > Thank your for your detail responsese.
> >
> > W1: I am satisfied with the answer. I hope the authors should consider the comparison with the mentioned reference [A], as equivariant stochastic models have been explored before.
> >
> > W2 and W3: I notice the new results on equivariant deterministic methods and the equivariant or invariant latent. That is great. However, I would expect more explanations on why stochastic methods are beneficial. The authors should explain more on this point since it is indeed the main motivation and contribution of this paper. As for the equivariant baselines, I understand that implementing equivariant ODEVAE could be difficult. But the authors should do more on taking other trivial equivariant probabilistic models into comparison.
> >
> > W4: Thank you.
> >
> > W5: The explanations should be included in the revised paper.
> >
> > W6: Thank you.
> >
> > W7: More details should be included in the main paper.
> >
> > Overall, the authors’ responses are helpful and positive. I am willing to increase my score. Given the technical novelty and contributions are not that significant, 5 is the best I can give.

---

> > > ### Author Response · Authors · 2023-08-18
> > > **Thank you for the response and suggestions**
> > >
> > > Thank you for responding to the rebuttal and increasing your score. We will take the suggestions of your review and comment into account when updating the paper.

---

### Official Review · Reviewer_n81N · 2023-07-07

**Soundness:** 4 excellent
**Presentation:** 4 excellent
**Contribution:** 3 good
**Rating:** 7
**Confidence:** 3

**Summary:**

This paper introduces a general framework named Equivariant Probabilistic Neural Simulation (EPNS) for equivariant stochastic simulation with neural simulators. The framework incorporates domain equivariances and utilizes an autoregressive probabilistic approach. The authors prove that, under equivariance assumptions on the neural network layers, the probability distribution over trajectories remains equivariant. The proposed framework is applied to two scenarios: a stochastic n-body system with celestial dynamics and stochastic cellular dynamics on a grid domain. Additionally, the authors propose a novel message passing graph neural architecture that maintains equivariance to permutations of cell indices, motivated by stochastic cellular dynamics problem. Experiments showcase multiple facets of the benefits of this framework.


**Strengths:**

S1. The paper presents a method for learning stochastic equivariant distributions over trajectories, which constitutes a valuable contribution to neural simulation. Additionally, the authors introduce a novel message passing graph neural network that enables equivariant embeddings in domains containing lattices and sets.

S2. The experiments are comprehensive and demonstrate several benefits of the framework, including improved data efficiency, uncertainty quantification, and rollout stability.

S3. The paper's presentation is excellent, with a detailed appendix and code that facilitates reproduction of results. The results are comprehensive, and the insights are well-described.

**Weaknesses:**

W1. The authors state the benefits of learning equivariant distributions over equivariant function approximations. However, it is unclear whether any of their experiments support this claim beyond enabling uncertainty quantification.

W2. The authors did not explicitly state whether the simulations used for the experiments are toy examples or scenarios that are of significant interest to domain experts. It would be beneficial for the paper to provide clarification on this aspect. Specifying the nature and relevance of the simulations would help us assess the potential impact of the developed methods on scientific applications. This information is crucial for understanding the current progress in developing methods that have practical implications in the scientific community.

**Questions:**

Q1.  While I understand how equivariance can help in maintaining cell clusters and representing the true number of clusters, I am unclear about its connection to kinetic energy in celestial dynamics. Could the authors provide further insight into how equivariance relates to kinetic energy in this context?

Q2. The authors did not cite any papers when selecting the metrics presented. Could the authors explain the rationale behind not basing their metric selection on previous work in neural simulation?

Q3. How can EPNS be extended to handle non-Markovian dynamics? What would be the potential benefits of such an extension?

Q4. Please comment to W2.

**Limitations:**

Limitations were mentioned. See W2 for another potential limitation.

---

> ### Author Rebuttal · Authors · 2023-08-09
>
> Thank you for the insightful review. We are happy to hear that you believe that EPNS is a valuable contribution to the field of neural simulation, that you find our experiments to be comprehensive, and that you liked the presentation of the paper. Please find our specific responses to your points below:
>
> **W1:** Initially, we took the need for a probabilistic model that is capable of sampling as a requirement to our problem setting of stochastic simulation, and aimed to demonstrate the benefits of equivariant distributions over non-equivariant counterparts. We have now run additional experiments with deterministic versions of the EPNS models on our datasets to investigate whether there are other benefits to using probabilistic models on these problem settings. The results can be found in Tables R1 and R2 and Figures 1a, 1c and 3b in the general response. Interestingly, using EPNS instead of its deterministic variant improves rollout stability for celestial dynamics; for cellular dynamics, stability for the deterministic model appears better at first glance, but this is only because the model produces ‘frozen’ dynamics, as shown in Figure 3b, which is not the case for the ground truth. To quantify this, we calculated the mean cell velocity, which is shown in Table R4. Of course, the D_KS metric, indicating calibration of model-sampled trajectories, is better in the probabilistic case.
>
> **W2:** Thank you for making us aware of this unclarity. We did not design our experiments with the aim of trying to solve open problems in two distinct domains. However, this does not mean that the experimental settings are easy from a machine learning modeling perspective, demonstrated by the clear performance difference between EPNS and the baseline methods for both scenarios. Additionally, for the n-body case, the complexity of the data (5 bodies, corresponding to a 30-dimensional SDE) is very similar to experiments in recent machine learning papers [1, 2, 3]. For the cellular dynamics case, the high dimensionality and level of complexity of the dynamics is comparable to computational biology studies that use the ground-truth simulation algorithm to investigate the migration of cancer [4, 5]. So, although these specific settings are not *directly* of specific interest in their relevant application domains, they are of representative complexity of problems studied in ML literature (for n-body dynamics) and in computational biology (for cellular dynamics simulations).
> We will update our paper to include an explicit discussion on the scope, relevance and limitations of the experimental setup.
>
> **Q1:** The E(n) symmetries introduced into the model architecture do not directly contribute to better modeling of quantities like kinetic energy in the n-body experiment. Nevertheless, for the n-body case, we observe that EPNS performs better than the non-equivariant models in terms of the evolution of the kinetic/potential energy of the system. The explanation for this is that including symmetries enables more effective and efficient learning of the dynamics in general, which is reflected in aggregate quantities like kinetic energy.
>
> **Q2:** Log-likelihood is a common metric for evaluating probabilistic models, and has also been used in previous works (e.g. [2]) – we will make sure to add relevant citations to emphasize this. In addition to evaluating goodness-of-fit, we want to evaluate whether our simulations are physically realistic. To do so, we look at characteristics of the underlying dynamics, such as various types of energy for the n-body system and number of cell clusters in the cellular dynamics, and compare the distributions of these to the ground truth. What characteristics are relevant strongly depends on the system at hand, meaning there is no one metric of physicality applicable to all datasets or dynamics. However, the metric we use for comparing the distributions of these characteristics, the Kolmogorov-Smirnov test statistic, is well understood and widely used. Note that in practice, when running simulations, one is often particularly interested in these kinds of characteristics, so they are crucial in assessing the performance of a simulation model.
>
> **Q3:** In order to extend EPNS to non-Markovian dynamics, we need to be able to pass information (possibly arbitrarily many) timesteps into the future. One way to do this would be to introduce a memory component into the model, for example similar to LSTMs. However, we need to make sure that this does not break the equivariance properties of EPNS. As such, the updating of this component should be done in a similar way in which the latent variables are constructed in EPNS, e.g. by having equivariant memory updates, and a decoder which is equivariant with respect to this memory cell as well. The proof of Lemma 1 would then remain similar. Other approaches could be investigated as well, e.g. using transformers to extract information from the prior sequence. Further research is needed to investigate what works best empirically.
> The benefits of such extensions would be that this will make EPNS suitable for modeling systems where the Markov property does not hold, e.g. in case of long-range dependencies in the data-generating process or when not all relevant information of the system is observed.
>
> **Q4:** See W2.
>
> [1] V. G. Satorras et al. “E (n) equivariant graph neural networks”, ICML, 2021
>
> [2] C. Yildiz et al. “Learning interacting dynamical systems
> with latent gaussian process ODEs”, NeurIPS, 2022
>
> [3] O. Puny et al. “Frame averaging for invariant and equivariant network design”, ICLR, 2022
>
> [4] S. Kumar et al. “Proteolytic and non-proteolytic regulation of collective cell invasion: tuning by ECM density and organization” Scientific Reports 6, 2016.
>
> [5] Y. Zhang et al. “Computational Modeling to Determine the Effect of Phenotypic Heterogeneity in Tumors on the Collective Tumor–Immune Interactions”. Bull Math Biol 85, 2023.

---

### Official Review · Reviewer_QKcR · 2023-07-08

**Soundness:** 3 good
**Presentation:** 3 good
**Contribution:** 3 good
**Rating:** 4
**Confidence:** 4

**Summary:**

This work presents a new framework for autoregressive probabilistic modeling called Equivariant Probabilistic Neural Simulation (EPNS) that produces equivariant distributions over system evolutions. The authors demonstrate that this method outperforms existing neural network-based methods for probabilistic simulation, improving simulation quality, data efficiency, rollout stability, and uncertainty quantification. The paper also discusses the benefits of incorporating domain symmetries in deterministic neural simulators and provides specific examples of domains where EPNS could be applied for efficient and effective data-driven probabilistic simulation.

**Strengths:**

Authors attempt to solve a challenging problem of simulating systems following a probabilistic simulation. They present an autoregressive approach toward probabilistic simulations combining equivariance. This can be used for several problems such as Langevin dynamics, Brownian motion to name a few.

**Weaknesses:**

There are several weaknesses for the work as outlined below.

1. **Simple systems:** Authors have considered very simple toy systems such as 3-body gravitational system, and a simplified cellular dynamics, a lattice based system. There are several examples of realistic probabilistic simulations such as Langevin or Brownian dynamics; these are not considered. This limits the application to toy systems and validity of such approach for larger realistic systems are questionable.
2. **Generalizability:** The work is demonstrated on a 3-body gravitational system. The GNN nature of the approach allows natural zero-generalizability to larger system sizes. These are not explored making the application of the approach to real-world problems limited. Similarly, due to the data-driven approach, it is not clear whether the approach can be generalized to unseen noise conditions for the same system. That is, for the gravitational system, if the interactions with the dust particles increased or for the cellular the systems, if the temperature increased, will the approach be applicable is not clear.
3. **Metrics:** This approach is presented as a data-driven approach to learn the dynamics. However, the ability of the model to preserve the physical quantities meaningfully is not explored. For instance, the forces on each particle in the rollout, how do they compare with those in the ground truth in a statistical fashion? Similarly, the phase space explored by the rollout, do they remain same ensuring statistical similarity of the trajectory. Indeed, the authors have shown the $LL$ and $D_{KS}$, but more elaborate error metrics can provide insights into the nature of learned simulations.
4. **Baselines:** It is not clear why a graph architecture is chosen especially when there is no generalizability to larger system sizes explored. To this extent, an additional MLP-based baseline can give insights into whether a GNN architecture provides superior results or not.

**Questions:**

There are several concerns regarding the work as outlined below.
1. There are several equivariant architectures including the EGNN authors have mentioned. The choice of the specific GNN which is not truly state-of-the-art for dynamics simulation is not clear. Authors mentioned that EGNN is not stable while training. EGNN has one of the simplest architectures among the equivariant GNNs and it is not clear why such an architecture would be unstable. Authors should clarify this.
2. The reason for the choice of a graph for 3-body system is not clear. Why not an MLP for such a simple system?
3. Can the graph architecture be generalized to larger systems, say 30 or 300 particles? It is suggested to demonstrate this ability. Otherwise, the approach remains applicable only to toy systems.
4. What are the characteristics of the learned dynamics? Can the drift and diffusion term of the SDE be decoupled? If yes, how well is the variance of the drift term captured by the simulator? If not, how can the ability of the model to realistically capture the simulation be established.
5. Authors discuss about the stability of the simulation. Indeed, autoregressive models are known to be unstable. Can the authors discuss about the stability of the learned simulation with respect to the length of the training trajectory or inference trajectory? That is, can they say that the inference trajectory will remain stable for the same length as the training trajectory or lower or more $x$ times more. This would be helpful to establish the usefulness of the ML-model
6. What about the data-efficiency of the approach? Does it continue to become better with increasing data. If yes, by how much? Have authors investigated the data efficiency of the model?
7. Finally, the model is not interpretable and not generalizable to larger system sizes. For instance, the model cannot give any insights about the physics or dynamics of the system in terms of the drift or the diffusion term. This means it can simulate the same system on which it is trained. This would be useful only if the model is computationally much more economical than the ground truth. Is that the case? Authors have mentioned that the GNN is computationally expensive. In that case, where do the authors think the approach would be useful practically?

**Limitations:**

While some of the limitations are briefly mentioned, there are a large number of limitations for the present work as discussed in the weaknesses and questions section. Authors should either address those or include those in the manuscript as limitations.

---

> ### Author Rebuttal · Authors · 2023-08-09
>
> Thank you for the review and the concrete comments and suggestions. Please find below our response to your points:
>
> **W1:** To avoid misunderstandings, let us clarify that the n-body experiments were conducted on a 5-body problem, rather than a 3-body problem. This means the data comes from a 30-dimensional SDE - it has three spatial coordinates and three velocity components for each of the five bodies. Moreover, similar 5-body system dynamics prediction problems are commonly used for evaluating new machine learning methods, see [1, 2, 3] for recent examples.
>
> For cellular dynamics, the data forms a system of 80x80 lattice sites, for each of which we need to produce a 64-dimensional probability vector for the cell index, as there are 64 cells which serve as nodes in the EPNS model. So, this system is extremely high-dimensional, and its complexity is evident from the poor performance of baseline models.
>
> **W2:** Let us first address generalizability to larger systems, also in response to Q3 and Q7. For cellular dynamics the system is already large, as there are n=64 cells on a 80x80 grid. For the n-body dynamics, we have now run additional experiments with n=20 bodies to demonstrate EPNS’ capability to generalize to larger system sizes – the ground-truth simulator struggled to remain stable for larger n. Results can be found in Table R3 and Figures 2 and 3a in the general response. The key observations are that performance for 20-body dynamics case is similar to the 5-body case, as evidenced by good match with the ground truth in the coordinate exploration plots (Figure 2, compare bottom and top rows), and the LL and D_KS scores (Table R3), which together with Figure 3a indicate the large overlap between EPNS’ trajectory distribution and the ground truth. In terms of generalizing to e.g. different noise conditions and temperatures, EPNS could easily be extended to take such variables as a conditioning signal, similar to e.g. [4], but this is left out of scope for this paper.
>
> **W3:** For the 5-body system (resp. for the cellular dynamics), we have investigated the behavior of physical quantities such as kinetic energy and potential energy (resp. the number of cell clusters and cell volume), as shown in Figures 5, 6 and Table 1 of our original paper and the accompanying text. However, the quantities you suggested are interesting to investigate as well. In terms of forces, due to the specific parameterization of our model (Appendix B.1), the forces are not directly modeled, so this is not possible. For the phase space exploration, we have added plots showing the distribution over the spatial coordinates for a body in Fig. 2 of the general response PDF – we will add more plots in the appendix of the updated version of the paper where we have more space. The probability distribution over coordinate values between the ground truth and EPNS overlaps very well, both for 5-body and 20-body systems (top and bottom rows of Fig. 2).
>
> **W4:** This work is about the incorporation of equivariances into stochastic simulation models. The 5-body system has a permutation symmetry - it does not matter for the physics in which order we label the planets - and a graph neural network is equivariant to precisely this symmetry, whereas an MLP is not. To stress the importance of the permutation symmetry, we have now run experiments with an MLP baseline. As can be seen in Figures 1b, 2, and Table R1 in the general response, this performs much worse than EPNS.
>
> **Q1:** The FA-GNN is SOTA for n-body dynamics prediction, see Section 5.3 of [3] for a comparison to EGNN. The instability of EGNN occurred in combination with the multi-step training heuristic, whereas the FA-GNN suffered significantly less from this problem. So, the choice for FA-GNN in this specific EPNS model is motivated both from a performance and practical perspective.
>
> **Q2:** See W4.
>
> **Q3:** See W2.
>
> **Q4:** Our method is not restricted to modeling SDEs - see for example the cellular dynamics application - and as such we do not model a separate drift and diffusion term, since those are only applicable to SDEs. The ability to realistically capture the dynamics can be established by looking at domain-specific quantities of interest such as kinetic and potential energy for celestial dynamics, the number of clusters for cellular dynamics, the phase space distributions you suggested, and common metrics like log-likelihood.
>
> **Q5:** The maximum rollout length for EPNS during training was 16 steps for the 5-body problem and 14 steps for the cellular dynamics. As can be seen from Figure 6 in the paper, EPNS generally remains stable for much longer than the rollouts seen during training.
>
> **Q6:** We have investigated data efficiency, see Table 3 in the paper and the accompanying text.
>
> **Q7:** See W2 and Q4 for our comments on larger systems and interpretability. Further, our primary goal was to demonstrate the potential of our method for effectively learning stochastic dynamics from data. Nevertheless, the n-body EPNS model is very economical in terms of simulation time: around 20 times faster than the ground truth simulator. Further, as can be seen in Table 3 of our paper, our method is data efficient, meaning that with few samples, a model can be trained that can simulate from many initial conditions.
> More importantly, this work constitutes a step in the direction of data-driven simulations of complex stochastic systems, which can be useful in many real-world settings where the true laws governing the dynamical system are unknown.
>
> [1] V. G. Satorras et al. “E(n) equivariant graph neural networks”, ICML 2021
>
> [2] C. Yildiz et al. “Learning interacting dynamical systems with latent gaussian process ODEs”, NeurIPS 2022
>
> [3] O. Puny et al. “Frame averaging for invariant and equivariant network design”, ICLR 2022
>
> [4] J. Brandstetter et al. “Message passing PDE solvers”, ICLR 2022

---

> > ### Comment · Reviewer_QKcR · 2023-08-20
> > **Thank you**
> >
> > Thank you for the detailed response. I would suggest authors modify the manuscript's limitations and future work aspects to reflect these discussions. I have raised my score.
> >
> > While I agree that the 20-body system is reasonable, the 80x80 grid doesn't truly represent a challenging high-dimensional problem. This is because a practical situation will not have a fixed 80x80 grid, and this volume will change. A problem of 6400 particles moving in space would be much more challenging than the 80x80 grid, although the dimensionality of each particle is only 3 and not 64. A realistic application of such stochastic simulators would require dealing much more particles (of the order of 100s or thousands) with significantly more complex interactions. Authors may include this in the limitations section if they agree.

---

> > > ### Author Response · Authors · 2023-08-20
> > > **Thank you for the response**
> > >
> > > Thank you for responding to the rebuttal and raising your score. You make a fair point, and we agree that including a discussion on this in the manuscript further improves the work -- thank you again for raising these points in your review. Please see our more detailed response below.
> > >
> > > For the cellular dynamics problem, the way of looking at it from an EPNS modeling perspective is that the system consists of 64 cells, each of which serves as a node in the GNN architecture. The dimensionality of each node would then be 80x80 lattice sites; of course, these sites are strongly correlated, so the underlying degrees of freedom for each node is lower, and this is precisely what is exploited very well by the proposed translation equivariant SpatialConv-GNN architecture. We will include a more precise discussion in the updated paper to clarify this explicitly.
> > >
> > > Furthermore, we recognize that scaling the model to a system size of the scale that you mention will introduce substantial challenges, and that this is an interesting avenue for further research. Consequently, we will extend the limitations section in the paper to reflect the points you made in your comment and review.
> > >
> > > Please do not hesitate to reach out if you have any further questions or comments. We believe that this discussion substantially improves the paper, and we hope you can agree!

---

### Official Review · Reviewer_AV4g · 2023-07-09

**Soundness:** 3 good
**Presentation:** 3 good
**Contribution:** 3 good
**Rating:** 6
**Confidence:** 3

**Summary:**

The paper presents EPNS, a framework for equivariance in autoregressive probabilistic multidimensional dynamics modeling. The framework is general such that one can use existing equivariant NN architectures; for the task of cellular dynamics forecasting the authors also introduce a ConvNet GNN. The models are shown to outperform current SOTA in both accuracy and data efficiency.

**Strengths:**

The framework and algorithm proposed is sound and general, which I can see being widely used in the spatiotemporal forecasting community. The authors did a great job outlining and explaining the EPNS algorithm step by step. The illustrations are great and helpful as well. The method even addressed the long rollout deviation problem common to autoregressive models. Overall, the algorithm is convincingly effective and practical.


**Weaknesses:**

- It was a little misleading when the authors say that they theoretically proved that EPNS produces equivariant distributions over trajectories. From my understanding, the equivariance comes from the paper's formulation by construction. Maybe a better way to word the contribution is something like "Our framework is able to construct equivariant distributions over the entire trajectory autoregressively from equivariant single-step predictions"?

- There is no substantial contributions on model architecture. I personally found the framework and training setup to be insightful enough to make an impact on the forecasting community, but one could argue it's just a new way to use existing models on new datasets.


**Questions:**

1. The paper mentions no work has considered equivariant probabilistic modeling of spatiotemporal dynamics. I can think of the following papers that tackles similar settings as in this paper:

- Köhler, Jonas, Leon Klein, and Frank Noé. "Equivariant flows: sampling configurations for multi-body systems with symmetric energies." arXiv preprint arXiv:1910.00753 (2019).
- Garcia Satorras, Victor, Emiel Hoogeboom, Fabian Fuchs, Ingmar Posner, and Max Welling. "E (n) equivariant normalizing flows." Advances in Neural Information Processing Systems 34 (2021): 4181-4192.
- Sun, Sophia Huiwen, Robin Walters, Jinxi Li, and Rose Yu. "Probabilistic Symmetry for Multi-Agent Dynamics." In Learning for Dynamics and Control Conference, pp. 1231-1244. PMLR, 2023.

I believe your framework is more general w.r.t. symmetry groups, but it would be good to cite these works as well to provide context.

2. How are the p-values in table 2 calculated, and how should we interpret them?

3. Some minor questions on writing:
-  Is it a typo on line 143? equivariant with regards to $h^t$?
- Line 277: "meaning that new simulations also match better to the ground truth in terms of the examined properties" <- what does 'new simulation' and 'examined properties' refer to here?




**Limitations:**

The author provides a detailed discussion of limitations in the conclusion section of the paper.

---

> ### Author Rebuttal · Authors · 2023-08-09
>
> Thank you for your valuable review and specific comments and pointers. We are glad that you found our work to be sound and general, convincingly effective and practical, and well-explained. Please find our responses to your specific comments below:
>
> **W1 - wording of contribution:** thank you for pointing out that this formulation can be confusing. We will change the formulation to ‘We demonstrate that EPNS produces single-step distributions that are equivariant to transformations of the relevant symmetry group, and that it employs these distributions to autoregressively construct equivariant distributions over the entire trajectory’.
>
> **W2 - contributions on model architecture:** While the primary focus of this paper is the general framework for creating and learning equivariant distributions over trajectories, we also introduced the novel SpatialConv-GNN architecture. This new GNN architecture was necessary in order to respect both the shift and permutation symmetries of the cellular dynamics problem, and is described in Section 4.2 and Appendix B.2. In our opinion, this constitutes a substantial architectural innovation, since it exemplifies designing an EPNS model tailored to a combination of data symmetries - in this case permutational and shift symmetry on a grid domain - that is previously unexplored. In other words, the value does not only come from being a new layer, but also from demonstrating how application-specific architectures can be designed that greatly outperform generic baselines using the EPNS framework.
>
> **Q1 - related literature:** Thank you for pointing us to these relevant works. While we briefly mentioned the first two papers in Section 2.1 (due to space constraints), we will extend the discussion on them in the updated version of the paper. Both of these works also use equivariant models in the parameterization of probability distributions. A crucial difference is that their time parameter is internal to the model and used to define a continuous flow from a base distribution to a target distribution, whereas we explicitly consider time as a variable in the data. Furthermore, they only consider modeling E(n)-invariant distributions, as you already mentioned.
> The third paper is more closely related to our work, and it was published in PMLR after the NeurIPS submission deadline. Nevertheless, we will discuss it in our updated related work section. The major differences are twofold:
> 1. generality of symmetries: the method in the third paper considers only planar rotations (SO(2)), while we consider general linear symmetry groups. Concretely, this means that their method is not applicable to e.g. the cellular dynamics problem or the three-dimensional n-body problem of our experimental evaluation;
> 2. model expressivity: the one-step distribution of their method is restricted to being a multivariate normal distribution. In contrast, by introducing a latent variable into the model architecture, our method can model more general one-step distributions. This should be especially relevant in the case the one-step distribution can no longer be assumed to be unimodal, for example due to coarse temporal resolution and the presence of bifurcations in the dynamics underlying the data.
>
> **Q2 - p-values:** the p-values in table 2 are calculated using the two-sample Kolmogorov-Smirnov (KS) test, a statistical test of equality of two probability distributions. The null hypothesis of this test is that the two sample sets under consideration are drawn from the same probability distribution. To test the null hypothesis, it compares the empirical cumulative distribution functions (CDFs) of the two sample sets under consideration, finds the maximum distance between them, and calculates a p-value based on this statistic. This p-value basically expresses the probability of finding a distance that is at least as large as the actual measured distance, under the assumption that the null hypothesis is true. As such, if the p-value is small (say, less than 0.05), it is very unlikely that we encounter this distance (or an even larger one) under the null hypothesis, and thus we reject it. For EPNS, the large p-values in Table 2 of our paper indicate that the KS test cannot distinguish between the distributions from which the two sets are sampled, meaning that the null hypothesis that the underlying distributions are statistically equivalent cannot be refuted. In contrast, for PNS, the p-values are very small, meaning that according to the KS test it is very unlikely that the underlying distributions are identical, providing strong evidence that the equivariance property is violated.
>
> **Q3 - typo:** Thank you for spotting this typo. Indeed that should have been “equivariant with regards to”.
>
> **Q3 - line 277:**  We now see that this formulation can be a bit confusing. With ‘new simulations’, we mean sampled trajectories from the ML model starting from the given initial condition x^0. This is in contrast to the log-likelihood, which is measured on ground-truth trajectories from the test set, i.e. no sampling from the model is involved.  With examined properties, we mean the properties of the system mentioned in the caption of Table 1, i.e. kinetic energy for the n-body system and the number of cell clusters of the same type for cellular dynamics, which are aggregate properties that are relevant to these specific domains.

---

> > ### Comment · Reviewer_AV4g · 2023-08-15
> >
> > Thank you for your thorough response. I have increased my score to reflect the author's edits and additions.

---

> > > ### Author Response · Authors · 2023-08-18
> > > **Thank you for the response**
> > >
> > > Thank you for reading the rebuttal and increasing your score. We will take your suggestions into account for the updated version of the paper.

---

### Author Rebuttal · Authors · 2023-08-09

# General response to reviewers

We would like to thank all the reviewers for their insightful and detailed comments! We are happy to read that the reviewers generally agree that our work tackles a valuable problem, that we present an interesting and effective framework and that the paper is well-written.

Two overarching points were mentioned multiple times, which are also addressed in more detail in the individual responses:
* The need for extra baselines/ablations to verify the utility of the stochastic modeling framework and the architectural design choices. We address this by providing new experiments with:
    + Deterministic equivariant methods (verifying the utility of stochastic modeling);
    + Modeling the transition probability directly without a latent variable (verifying the utility of the latent-variable approach);
    + Using equivariant (resp. invariant) latents for the celestial dynamics (resp. cellular dynamics) rather than the setting used in the paper (ablation study of design choices);
    + Using a non-equivariant baseline (MLP) for celestial dynamics (validating the equivariant/GNN approach).
* The complexity of the considered systems, and a discussion thereof. We address this by elaborating on the systems in more detail, and we kindly wish to highlight recent works addressing very similar systems [1, 2, 3]. Further, the complexity is also demonstrated through the baseline methods struggling in some scenarios. Finally, to better demonstrate the scalability of our approach for the n-body case, we have performed additional experiments on a system of 20 bodies, as opposed to 5 bodies in the paper.

In addition, we address individual comments with respect to our work in the context of other related works, and explain how the paper will be updated to include a discussion on these works. We thank the reviewers for pointing us to these references. Further, we have clarified some specific pieces of the paper that could be confusing or misinterpreted. Finally, we have addressed the remaining specific individual questions.

The additional results can be found in the tables at the bottom of this general response and in the figures in the attached pdf file, and are further explained in the individual responses.

We would like to again thank the reviewers for their detailed comments as it allowed us to substantially improve our work!

[1] V. G. Satorras et al. “E(n) equivariant graph neural networks”, ICML 2021

[2] C. Yildiz et al. “Learning interacting dynamical systems with latent gaussian process ODEs”, NeurIPS 2022

[3] O. Puny et al. “Frame averaging for invariant and equivariant network design”, ICLR 2022

---

Table R1: additional results of ablations for celestial dynamics.
|  | NSDE | iGPODE | PNS | EPNS |\||  EPNS-equivariant latents | EPNS-no latents | EPNS-deterministic | PNS-MLP |
|-|:-:|:-:|:-:|:-:|-|:-:|:-:|:-:|:-:|
| $\uparrow$ LL ($\cdot 10^3$) |-5.0±0.0|-8.5±0.1|11.3±0.3|11.6±0.1|\||10.9±0.1|**13.2±0.1**|N/A|5.9±0.3|
| $\downarrow$ $D_{ks}(KE)$ t=50 |0.80±0.01|0.42±0.07|0.55±0.13|0.39±0.12|\||**0.13±0.03**|0.25±0.12|0.63±0.08|0.52±0.21|
| $\downarrow$ $D_{ks}(KE)$ t=100 |0.65±0.03|0.57±0.02|0.30±0.10|**0.18±0.02**|\||**0.18±0.02**|**0.19±0.05**|0.7±0.05|0.44±0.11|

---

Table R2: additional results for ablations for cellular dynamics
|  | ODE$^2$VAE| PNS | EPNS |\||  EPNS-invariant latent | EPNS-no latents | EPNS-deterministic |
|-|:-:|:-:|:-:|-|:-:|:-:|:-:|
| $\uparrow$ LL ($\cdot 10^4$) |-30.1±0.0|-16.4±0.3|**-5.9±0.1**|\||-58.2|-9.77|N/A|
| $\downarrow$ $D_{ks}$(#clusters) t=30 |0.98±0.01|0.70±0.10|**0.58±0.09**|\||1|1|0.77|
| $\downarrow$ $D_{ks}$(#clusters) t=45|0.96±0.04|0.77±0.05|**0.58±0.05**|\||1|1|0.9|

---

Table R3: results for celestial dynamics with 20 bodies.
|  | PNS | EPNS |
|-|:-:|:-:|
| $\uparrow$ LL ($\cdot 10^3$) |15.3±0.2|15.7±0.0|
| $\downarrow$ $D_{ks}(KE)$; t=50 |0.54±0.29|0.28±0.01|
| $\downarrow$ $D_{ks}(KE)$; t=100 |0.19±0.02|0.26±0.12|

---

Table R4: mean velocity of cells (in #grid sites per step)
|Ground truth  | EPNS | EPNS - deterministic |
|-|:-:|:-:|
| 1.821 |1.35|0.125|

---

### Decision · Program_Chairs · 2023-09-21

**Decision:**

Accept (poster)

**Comment:**

The paper proposes a model for equivariance in distributions of trajectories rather than function approximators directly, the idea being that with stochastic simulators probabilistic models and distributions are the natural fit. All but one reviewer have a positive outlook to the paper. They appreciate the innovation, the generality, and good results in a wide array of benchmarks.

One of the reviewers voiced concerns regarding the scalability and generalizability of the method on higher-dimensional settings. I agree with the reviewer: the experiments are on rather small settings, and I do not find very convincing the argument that the 80x80x64 cellular dynamics problem constitutes a high-dimensional problem, because the dimensions here is due to the way the problem is parameterized for the sake of the approach. The question is whether the model can handle large systems (which have large dimensionalities), not whether they can handle any large dimensionality problem (one can also append 0s or random noise dimensions to trivially make very larger dimensional problems, this does not increase the intrinsic dimensionality, however).

That said, I agree with the sentiment of the authors, that there has been sufficient empirical justification and even they tried their method in a 20-body problem. Larger systems are for sure of interest, however, I think for this paper the novelty and validation is ok. I ask from the authors however to incorporate all the criticisms regarding the absence of truly high-dimensional systems and other types of dynamics (eg Langevin) in their limitations.